# A toll-like receptor agonist mimicking microbial signal to generate tumor-suppressive macrophages

Yanxian Feng [1,3], Ruoyu Mu[1,3], Zhenzhen Wang[2], Panfei Xing[1], Junfeng Zhang[2], Lei Dong[2] & Chunming Wang [1]

Switching macrophages from a pro-tumor type to an anti-tumor state is a promising strategy for cancer immunotherapy. Existing agents, many derived from bacterial components, have safety or specificity concerns. Here, we postulate that the structures of the bacterial signals can be mimicked by using non-toxic biomolecules of simple design. Based on bioactivity screening, we devise a glucomannan polysaccharide with acetyl modification at a degree of 1.8 (acGM-1.8), which specifically activates toll-like receptor 2 (TLR2) signaling and consequently induces macrophages into an anti-tumor phenotype. For acGM-1.8, the degree of acetyl modification, glucomannan pattern, and acetylation-induced assembly are three crucial factors for its bioactivity. In mice, intratumoral injection of acGM-1.8 suppresses the growth of two tumor models, and this polysaccharide demonstrates higher safety than four classical TLR agonists. In summary, we report the design of a new, safe, and specific TLR2 agonist that can generate macrophages with strong anti-tumor potential in mice.

[1] State Key Laboratory of Quality Research in Chinese Medicine, Institute of Chinese Medical Sciences, University of Macau, Taipa, Macau SAR, China. [2] State Key Laboratory of Pharmaceutical Biotechnology, School of Life Sciences, Nanjing University, Nanjing 210093, China. [3]These authors contributed equally: Yanxian Feng, Ruoyu Mu. Correspondence and requests for materials should be addressed to L.D. (email: leidong@nju.edu.cn) or to C.W. (email: cmwang@umac.mo)

A major goal in cancer immunotherapy is to switch tumor-associated macrophages from a pro-tumor to an anti-tumor state[1,2]. These cells act as a powerhouse for tumor angiogenesis and metastasis, by displaying an immunosuppressive (M2) phenotype and secreting abundant pro-tumor cytokines[3]. They play a pivotal role in activating both innate and adaptive immunity against the tumor[4], and should be switched to an immunostimulatory (M1) state to release anti-tumor cytokines. To realize this goal, a possible strategy is to design therapeutic agents that mimic the way microbial signals stimulate the immune cells[5–7]. For instance, certain bacterial and fungal structures, known as pathogen-associated molecular patterns (PAMPs), can activate the toll-like receptors (TLRs) on macrophage surface and stimulate the cells into a solid M1 phenotype[8,9]. Existing microbe-derived, TLR-targeting substance/agents range from the deactivated bacteria (Coley's toxin)[10], lipopolysaccharide (LPS, a cell-wall component of gram-negative bacteria; agonist of TLR4)[11], monophosphoryl lipid A (MPLA, a further derivative of LPS; agonist of TLR4) to $Pam_3CSK_4$ (a synthetic mimetic of bacterial lipopeptide; agonist of TLR1/2)[12]. However, their application in cancer immunotherapy is limited, because the bacterial derivatives are often too toxic and heterogenic, while $Pam_3CSK_4$ has both inconsistent anti-tumor efficacy and unconvincing in vivo safety[13–15]. A new, safe agent that can specifically instruct macrophages to perform anti-tumor functions remains highly demanded.

Revisiting the PAMP structures reveals that polysaccharides and aliphatic groups are typically presented[16,17]. The polysaccharide usually contains the units of glucans and/or mannans, which can be recognized by macrophage carbohydrate receptors including TLRs and c-type lectins[18–20]. The hydrophilic polysaccharide chains are often attached with hydrophobic, aliphatic groups, which play vital roles in exerting immunoactivity[21,22]. Also, the size of the signaling structure is crucial for its bioactivity[23,24], and one in branched or particulate forms is often more potent than the same carbohydrate in linear or soluble form[25,26]. Therefore, we hypothesize that a glucomannan (GM) polysaccharide modified with acetyl groups (acGM) represents the essential, PAMP-mimicking structure to generate macrophages with anti-tumor activities. In this design, GM offers repeating units of glucose and mannose, and acetylation adds the simplest possible aliphatic group to the sugar ring. TLRs are known to respond to di-/tri-acetyl groups when recognizing fungi[27].

Therefore, in this study, we synthesize acGM with a range of acetylation degrees and examine their effect on macrophage phenotypes. Our data show that, when the acetylation degree increases to 1.8, acGM assembles from linear molecules and exerts the activity to induce macrophages into a proinflammatory state (Fig. 1). Using microarray analysis, transgenic mice/cells, and tumor models, we identify acGM-1.8 as a new, specific, and safe TLR-2 agonist with the potential to regulate both innate and adaptive anti-cancer immune responses in vivo.

## Results

**Preparation of acGM with different acetylation degrees.** We prepared acGM samples with six different degrees of substitution (DS; from 0.1, 0.2, 0.6, 1.2, 1.8, to 3.0) via a one-step reaction (Supplementary Fig. 1a). A konjac-derived GM, with a molecular weight of 100 kDa and a natural acetylation DS of 0.2, served as the starting material (acGM-0.2). We reacted acGM-0.2 with pyridine/acetic anhydride mixture for different time to obtain acGM-0.6, −1.2, −1.8, and 3.0, or through a deacetylation process to prepare acGM-0.1 (Supplementary Fig. 1b). All the samples were characterized with IR (–C = O, carbonyl group at

$1735 \ cm^{-1}$; Supplementary Fig. 1c) and $^1H$ NMR ($\delta = 2.1$ ppm indicating hydrogen in acetyl group; Supplementary Fig. 1d). Increasing DS of acetylation led to higher hydrophobicity, which was confirmed by the contact angle measurement (Supplementary Fig. 1e).

We speculated that this increased hydrophobicity led to changes in the morphology of the polymer, which was confirmed by TEM observation. Indeed, as DS increased from 0.1 to 1.2, acGM gradually changed the morphology from dispersed threads to assembled particles; when DS reached 1.8, the samples exhibited a regular spheroidal shape with a homogenous diameter of 200–300 nanometers (Fig. 2a). This intriguing finding indicates that increased acetylation possibly induces the assembly of the GM polysaccharide.

**acGM-1.8 specifically stimulates macrophages into a M1 phenotype.** The morphological change generates new bioactivity – acGM-1.8 remarkably stimulated macrophages towards the M1 phenotype, while other acGM samples with lower DS could not. The ELISA results suggested that acGM-1.8 markedly upregulated the secretion of typical proinflammatory cytokines, TNF-α and IL-12 p70, in primary bone marrow-derived macrophages (BMDMs) – as potently as LPS did, by 14.1 and 8.3 folds compared with the PBS control (Fig. 2b). However, acGM with a DS lower than 1.2 had little effect and acGM with a higher DS (acGM-3.0) did not outperform acGM-1.8. Meanwhile, acGM-1.8 downregulated the secretion of typical anti-inflammatory cytokines IL-10 and TGF-β1, by 52 and 61%, respectively, compared with the control (Fig. 2c). Additionally, PCR analysis showed the same trend with the ELISA outcomes that acGM-1.8 increased the expression of M1 cytokines (Supplementary Fig. 2a) while decreasing that of M2 cytokines (Supplementary Fig. 2b) in BMDMs. We also analyzed the change of a M1 specific marker, CD11c, of BMDMs with flow cytometry (Fig. 2d), and observed that the expression of CD11c increased after acGM-1.8 stimulation – up to 73.8%, which was similar to that of LPS (100 ng/mL).

The above data showed that acGM-1.8 could stimulate unpolarized macrophages to a M1 phenotype; however, the tumor-associated macrophages are predominantly in a M2 state. Thus, we asked whether acGM-1.8 was potent enough to reverse M2 macrophages into a M1 phenotype. The BMDMs were stimulated with IL-4 and IL-13 into the M2 phenotype and treated with acGM-1.8 (100 µg/mL) for another 24 h. The outcomes from flow cytometry demonstrated that the level of two specific M2 markers, CD206 (Fig. 2e) and CD163 (Supplementary Fig. 2c), was reduced from 70.8 to 22.4% and 62.1 to 27.2%, respectively. Consistently, the ELISA data (Fig. 2f and g) showed that acGM-1.8, but not acGM-0.2, re-educated the pre-induced M2-type BMDMs to a M1 state, with significantly increased production of proinflammatory cytokines (TNF-α and IL-12, p70; Fig. 2f) and decreased expression of anti-inflammatory ones (TGF-β1, IL-10 and VEGF-A; Fig. 2g).

**Both "ac" and "GM" are important for the activity of acGM-1.8.** First, to validate the importance of acetylation for the macrophage-stimulatory activity of acGM-1.8, we prepared a series of GM esters by substituting the hydroxyl groups on the sugar chain with butyryl, hexanoyl, octanoyl, and decanoyl groups (-$CO[CH_2]_nCH_3$, where $n = 2, 4, 6$, and 8; in comparison with acGM where $n = 0$; Fig. 3a). The products were characterized by $^1H$ NMR (600 Hz, $CDCl_3$): $\delta$ 1.25 (–$CH_3$), $\delta$ 1.86, 2.34 (–$CH_2$) and $\delta$ 3.8–5.4 (–CH in carbohydrate) for –$CO[CH_2]_nCH_3$, where $n = 2, 4, 6$, and 8 (Supplementary Fig. 3a). As the aliphatic chain becomes longer, the modified GM shows increasing size and irregular/heterogenous morphology but little

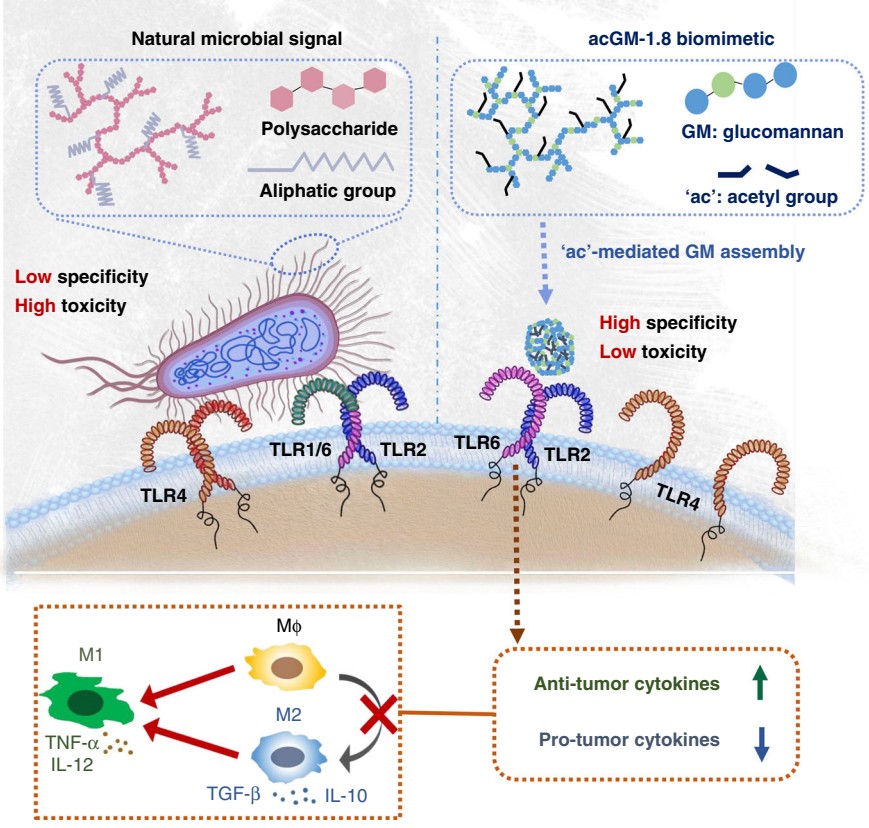

**Fig. 1** Schematic illustration of the design of acGM-1.8. An acetyl glucomannan polymer that can mimic microbial signals to stimulate macrophages to produce anti-tumor cytokines in a specific and non-toxic manner

change in zeta potential (Fig. 3b and Supplementary Fig. 3b). Among all these samples, only acGM exhibited a solid activity in stimulating macrophages towards M1 polarization. In all other samples, only buGM ($n = 2$) could moderately induce TNF-$\alpha$ and IL-12 p70 production, yet its effect was much weaker than that of acGM (Fig. 3c: ELISA; Supplementary Fig. 4a: RT-PCR). Similarly, none of these samples except acGM effectively lowered M2 gene expression (Fig. 3d: ELISA for TGF-$\beta$1 and IL-10; Supplementary Fig. 4b: RT-PCR for Arg-1, TGF-$\beta$1, and IL-10). These data highlight the crucial role of acetylation in the activity of acGM-1.8, which is not easily replaced by other modes of aliphatic modification.

Second, we asked whether the GM structure is vital to acGM-1.8's activity. We prepared acetyl dextran (acDex) with a comparable DS (1.9; Supplementary Fig. 5a) and tested its ability to stimulate macrophages. Nevertheless, acDex-1.9 was much less potent than acGM-1.8 in inducing the M0-to-M1 switch of BMDMs. The former only increased the level of one proinflammatory cytokine (TNF-$\alpha$; by 4.5 folds) and failed to decrease the expression of the anti-inflammatory cytokines (Fig. 3e); moreover, it was completely incapable of inducing the pre-induced M2-type BMDMs towards a M1 polarization (Fig. 3f). These findings suggest that the polysaccharide pattern of GM is also important for the activity of acGM-1.8.

Third, we investigated whether the assembly of acGM-1.8 is required for its activity. We treated acGM-1.8 with alkaline to break the hydrophobic interactions that probably underpin the assembly. Our finding showed that the size of acGM-1.8 was affected by pH (Supplementary Fig. 5b): when the pH decreased from 5.0 to 3.0, the size increased; but when the pH increased from 7.0 to 8.0 and finally to 10.0, acGM-1.8 exhibited a smaller size, and the suspension became a clear solution (Fig. 3g). We

performed IR spectrum to confirm that the alkaline treatment did not change the acetylation degree. Nevertheless, as the pH increased, the macrophage-activating effects of these disassembled acGM-1.8 samples were gradually weakened – though not completely abolished (Fig. 3h). These data suggest that, on top of the two essential structural features of ac and GM, the assembly of GM also significantly influences the bioactivity. This also explains why a sufficient degree of acetylation (i.e. 1.8) is required.

Together, these data provided essential information on the macrophage-stimulatory activity of acGM-1.8. First, acetylation, to an adequate degree, is necessary. Second, GM, the sugar unit, is equally crucial. Third, the nanoscale assembly of acGM to a proper size is also important. These three factors underpin the macrophage-regulatory activity of acGM-1.8.

**acGM-1.8 specifically activates the TLR2 signaling**. To understand the mechanisms of acGM-1.8 stimulating macrophages towards M1 polarization, we carried out a series of investigations. First, we performed microarray and ontology analysis to identify possible signaling pathways involved. The outcomes revealed that acGM-1.8 changed more proteomes than acGM-0.2 did (937 vs. 549, Fig. 4a). The expression of many proinflammatory genes was significantly increased in BMDMs treated with acGM-1.8, including IL-1$\beta$, IL-6, IL-12 p40 and iNOS; while that of key anti-inflammatory genes was suppressed, such as IL-10, IL-4, IL-13, and VEGF-A (Fig. 4b). These data, consistent with the ELISA and PCR outcomes presented above, confirmed the efficacy of acGM-1.8 (but not acGM-0.2) in stimulating macrophages into an M1 phenotype. Then, we compared the genes, whose expression was upregulated by both acGM-1.8 and acGM-0.2, into different

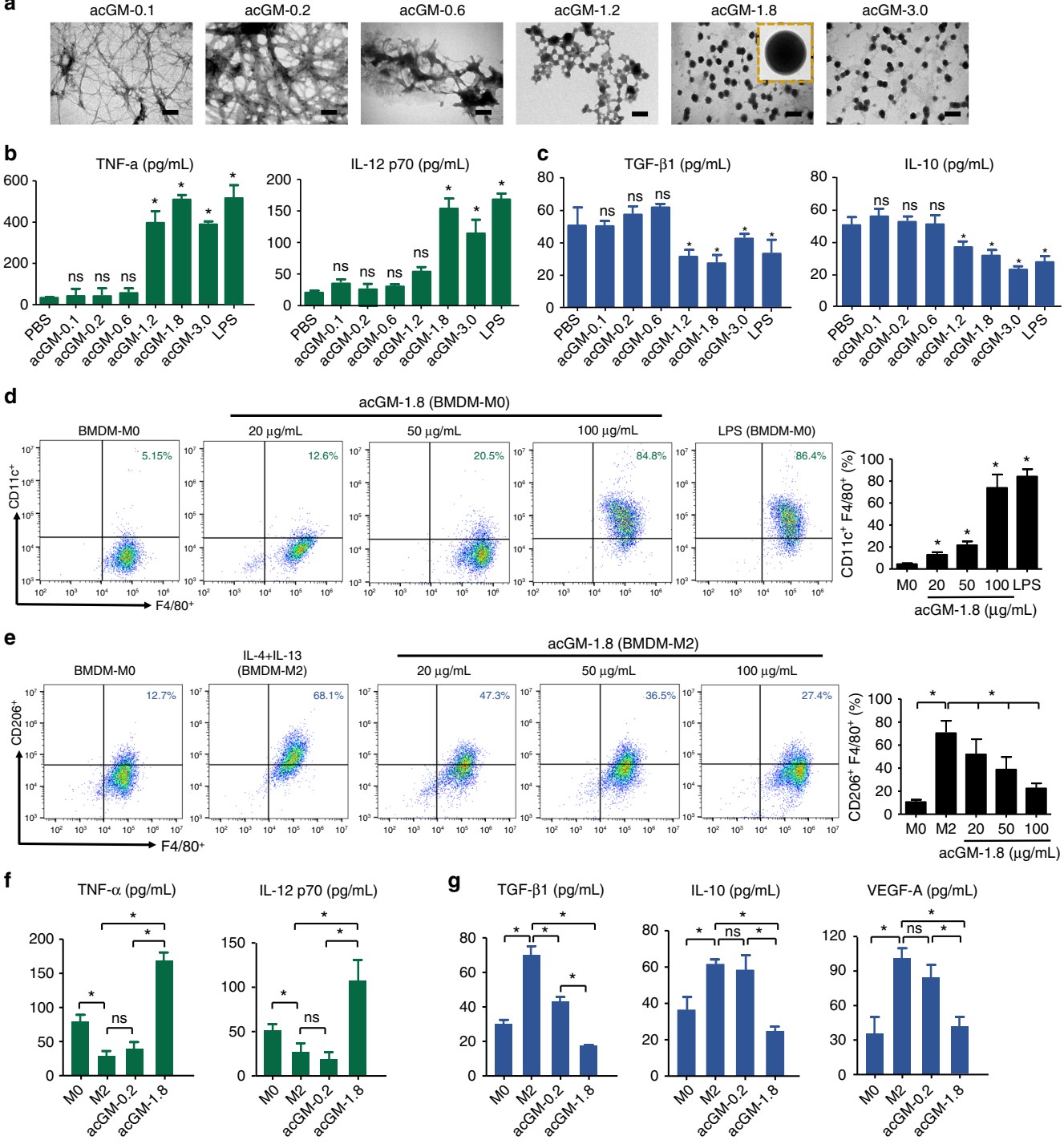

**Fig. 2** acGM-1.8 stimulates macrophages into a proinflammatory phenotype. **a** Representative TEM images of a series of acetyl glucomannan (acGM) samples with different degrees of acetylation (scale bar: 500 nm). **b** Determination of proinflammatory cytokines – tumor necrosis factor-α (TNF-α) and interleukin-12 (IL-12 p70) and (**c**) anti-inflammatory cytokines – transforming growth factor-β1 (TGF-β1) and interleukin-10 (IL-10) – secreted by primary murine bone marrow-derived macrophages (BMDM) after 24 h of stimulation by acGM samples (*$P < 0.05$ vs. the phosphate buffer saline [PBS] group; ns: no significance; $n = 3$). **d** Flow cytometry analysis of CD11c in BMDM after stimulation with acGM-1.8 (20, 50, and 100 μg/mL; *$P < 0.05$ versus the BMDM-M0 group; ns: no significance; $n = 3$). **e** Flow cytometry analysis of CD206 in BMDM that were pre-induced into M2 phenotype and then stimulated with acGM-1.8 (20, 50, and 100 μg/mL; *$P < 0.05$ versus the BMDM-M2 group; ns: no significance; $n = 3$). **f, g** ELISA analysis of (**f**) proinflammatory (TNF-α and IL-12 p70) and (**g**) anti-inflammatory cytokines (TGF-β1, IL-10, and VEGF-A) expressed by BMDM that were without treatment, induced into the M2 phenotype, and induced into the M2 phenotype followed by acGM-1.8/0.2 treatment (*$P < 0.05$ versus BMDM in the M2 phenotype; ns: no significance; $n = 3$). The data are representative for three independent experiments

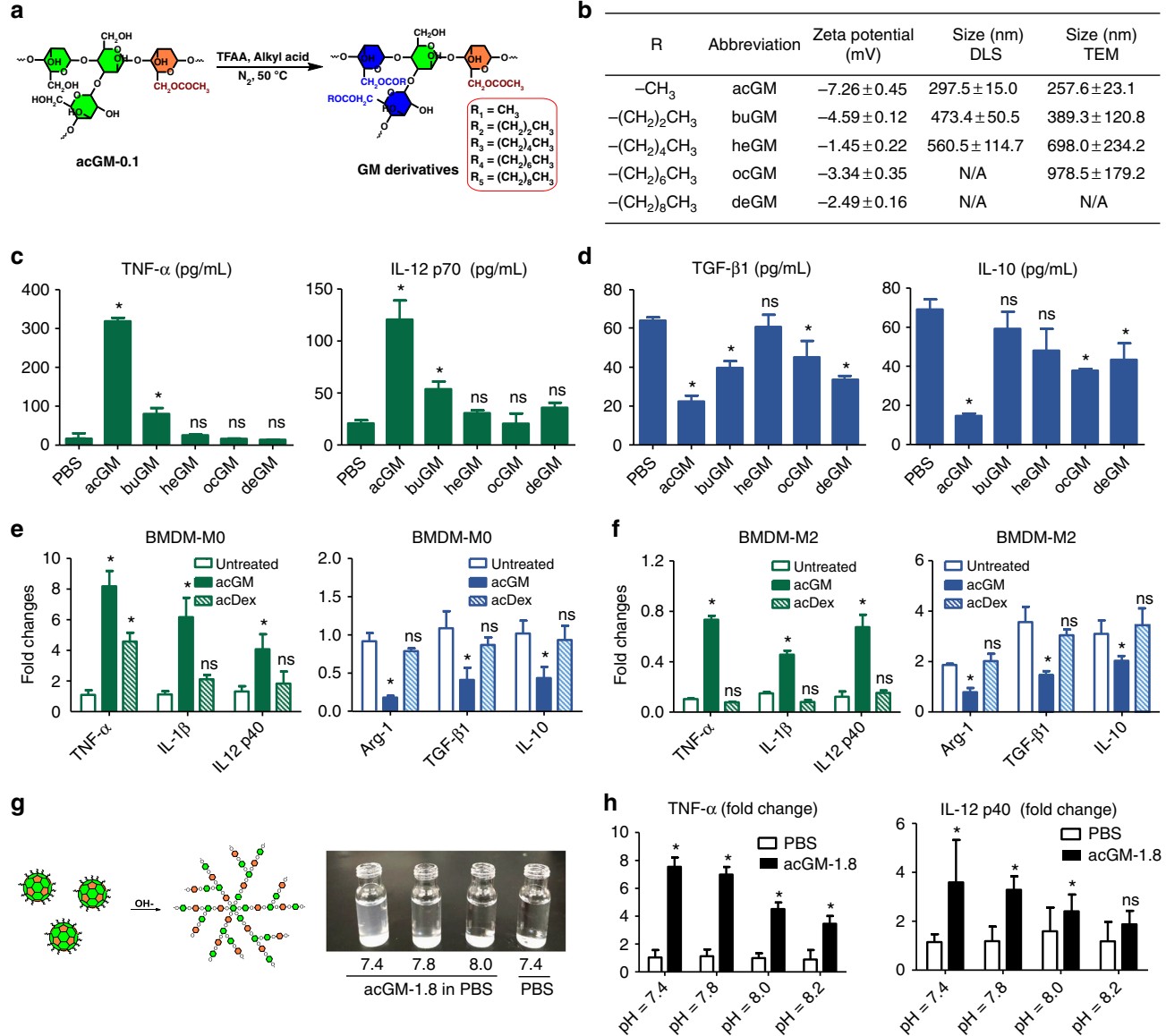

**Fig. 3** Both "ac" and "GM" are essential for the function of acGM-1.8. **a** Schematic illustration of preparing different GM derivatives. **b** Physical properties of the GM derivatives, including the type of aliphatic chain, zeta potential, and size determined by both dynamic light scattering (DLS) and transmission electron microscopy (TEM). **c**, **d** Determination of (**c**) proinflammatory cytokines TNF-α and IL-12 p70 and (**d**) anti-inflammatory cytokines TGF-β1 and IL-10 secreted by BMDM after 24 h of stimulation by various GM derivatives (*$P < 0.05$ vs. the PBS group; $n = 3$). **e** Fold changes of markers for macrophage polarization (M1: TNF-α, IL-1β, and IL-12 p40; M2: Arg-1, TGF-β1, and IL-10) in BMDM (BMDM-M0) treated with PBS, acGM, or acetyl dextran (acDex) for 24 h (*$P < 0.05$ versus the PBS group; ns: no significance; $n = 3$). **f** Fold changes of markers for macrophage polarization (M1: TNF-α, IL-1β, and IL-12 p40; M2: Arg-1, TGF-β1, and IL-10) in pre-induced M2-type BMDM (BMDM-M2) treated with PBS, acGM, or acetyl dextran (acDex) for 24 h. (*$P < 0.05$ vs. the PBS group; ns: no significance; the fold changes were normalized to those of M0; $n = 3$). **g** (left) Schematic illustration of breaking the acGM assemblies by alkaline solution and (right) the appearance of acGM-1.8 in PBS under different pH values (7.4, 7.8, and 8.0) compared with PBS. **h** Fold changes of proinflammatory cytokines (TNF-α and IL-12 p40) expressed by BMDM after incubation with PBS or acGM-1.8 under different pH values (7.4, 7.8, and 8.0; *$P < 0.05$ versus the PBS group; ns: no significance; $n = 3$). Data are representative for three independent experiments; ns: no significance

functional clusters relating to PRR or inflammation. We found that the biggest difference occurred to the genes involved in TLR pathway (56 genes in acGM-1.8 vs. 24 genes in acGM-0.2, Fig. 4c). The expression of several genes that play key roles in TLR signaling was significantly upregulated, including Irf7, Traf3, Traf6, Irak4 and Tollip, by 10.5, 5.6, 3.2, 3.0, and 2.4 folds, respectively (Supplementary Fig. 6a). Thus, we postulated that acGM-1.8 activated TLR signaling in stimulating the macrophages.

Next, to identify the specific TLR receptor for acGM-1.8, we employed both reporter cells and knockout mice models for TLR2 and TLR4 – the two major types of TLR on the cell membrane mediating proinflammatory activities. Pam₃CSK₄ and LPS (typical activator of TLR2 and TLR4, respectively) served as positive control. First, in the reporter cells, acGM-1.8, but not acGM-0.2, induced a strong response in the TLR2 reporter cell line, which was comparable to that triggered by Pam₃CSK₄ (Fig. 4d). However, both samples failed to generate signals in

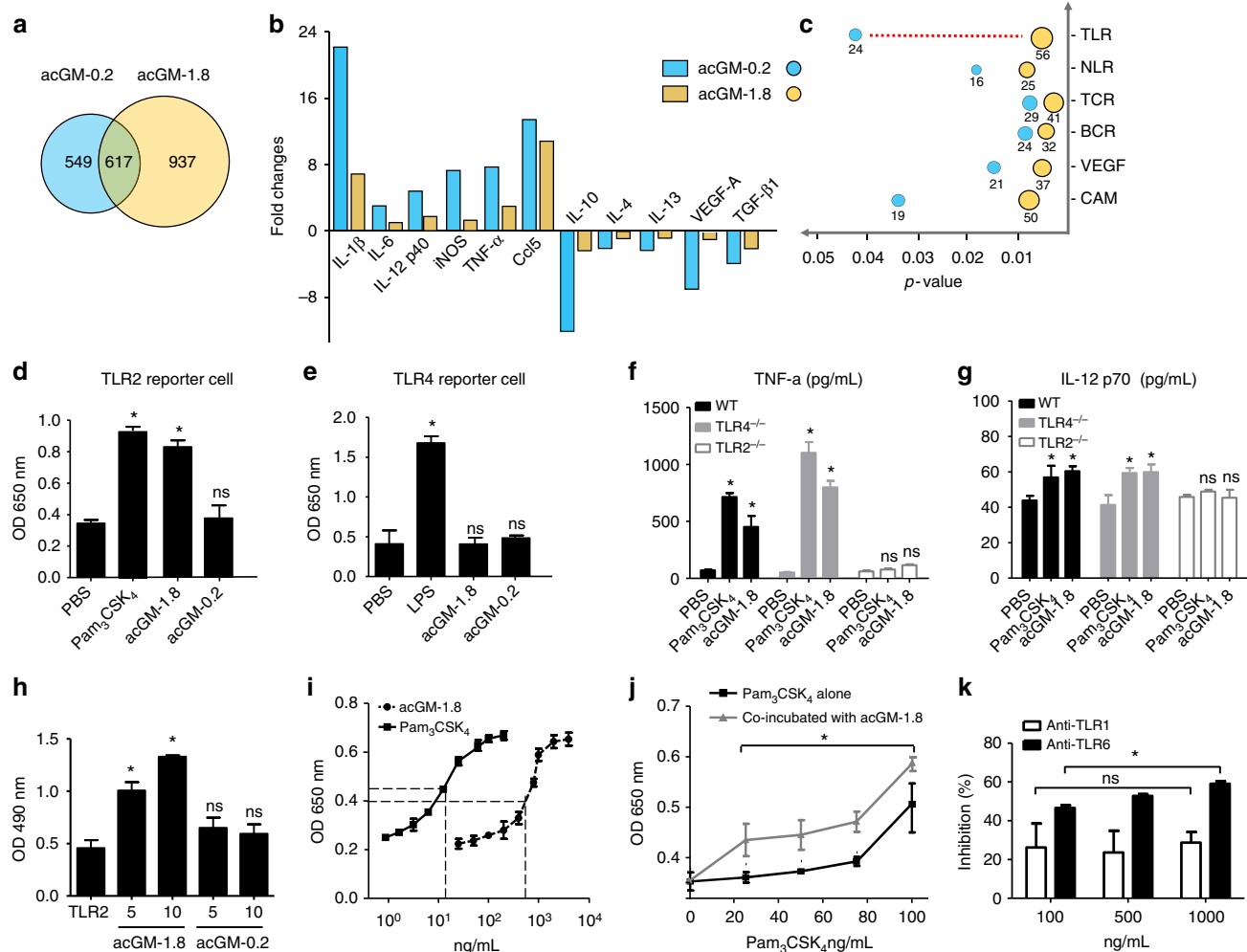

**Fig. 4** acGM1.8 specifically activates TLR2 signaling. **a–c** Microarray data on the gene expression in BMDM treated with acGM-1.8 or acGM-0.2: **a** Venn diagram of the gene numbers, (**b**) fold change of the markers related to macrophage polarization, and (**c**) top six signaling pathways changed after the treatment, where the numbers under the circles represent counts of different genes and the toll-like receptor (TLR) signaling pathway shows the greatest change. **d**, **e** Detection of secreted embryonic alkaline phosphatase (SEAP) activity in (**d**) TLR2 and (**e**) TLR4 reporter macrophages after the treatment of PBS, acGM-1.8, acGM-0.2, or the corresponding agonist. **f**, **g** Determination of cytokines (**f**) TNF-α and (**g**) IL-12 p70 secreted by macrophages of TLR-knockout mice or wild type mice (*$P < 0.05$ vs. the PBS group; $n = 3$). **h** Detection of acGM-1.8 or acGM-0.2 bound to TLR2: the TLR2 protein was pulled down and the bound polysaccharides were detected by sulfuric-phenol assay (*$P < 0.05$ versus the group of pure TLR2 without polysaccharides; $n = 3$). **i** Measurement of the $EC_{50}$ value of acGM-1.8: the TLR2 HEK-Blue cells were incubated with acGM-1.8 or Pam_3CSK_4 for 24 h, and the level of SEAP was recorded. **j** Assessment of potential competition between acGM-1.8 and the classical TLR2 agonist Pam_3CSK_4: the TLR2 HEK-Blue cells were treated with Pam_3CSK_4 alone (1 to 100 ng/mL) or Pam_3CSK_4 (1 to 100 ng/mL) along with acGM-1.8 (fixed at 1 μg/mL), and the level of SEAP was recorded. *$P < 0.05$ between the two compared groups; **k** Assessment of the role of TLR1 or TLR6 in acGM-1.8's activation of TLR2: the TLR2 HEK-Blue cells were pre-incubated with anti-TLR1 or anti-TLR6 before the addition of acGM-1.8 (100 μg/mL). The culture medium was collected and the level of SEAP was recorded. For (**k**), *$P < 0.05$ between two inhibition at 100 ng/mL and 1000 ng/mL, $n = 3$. Data are representative for three independent experiments, except for the microarray data representing two independent biological samples. The original data for (**a–c**) are submitted to GEO; ns: no significance

TLR4 reporter cells (Fig. 4e), suggesting that acGM-1.8 activates TLR2 instead of TLR4 signaling.

Then, to confirm the specific activation of TLR2 by acGM-1.8, we applied it to treat peritoneal macrophages harvested from the TLR2$^{-/-}$ and TLR4$^{-/-}$ knockout (KO) mice. The ELISA data on TNF-α (Fig. 4f), IL-12 p70 (Fig. 4g), IL-1β (Supplementary Fig. 6b), and IL-10 (Supplementary Fig. 6c) indicated that acGM-1.8 lost its M1-towards stimulatory activity to TLR2$^{-/-}$ macrophages but preserved the effect on TLR4$^{-/-}$ ones. The results further suggest that acGM-1.8 specifically activates TLR2, instead of TLR4.

To continue, we examined the direct binding between TLR2 and acGM-1.8. First, we incubated acGM-1.8, −1.2, −0.6, or −0.2 (10 mg/mL) with the membrane proteins isolated from the lysate

of TLR2 reporter cells, followed by Western blotting that detected the presence of TLR2 in the elution of acGM-1.8 but not in that of acGM-0.2 or −0.6, indicating the binding between acGM-1.8 and TLR2 (Supplementary Fig. 7a). Second and oppositely, through the same process of cell incubation, we performed co-immunoprecipitation (co-IP) to pull down TLR2 and detected for polysaccharide (acGM-1.8 and −0.2; 5 or 10 mg/mL) bound to TLR2. The outcome from phenol-sulfuric acid staining assay (Supplementary Fig. 7b) and the quantification based on colorimetric absorbance (Fig. 4h) showed a much stronger signal in the acGM-1.8 samples, which double confirmed the presence of acGM-1.8 bound to TLR2.

Further, we determined the efficiency of acGM-1.8 in activating the TLR2 reporter cells. The data collected at 24 h revealed the

$EC_{50}$ value of acGM-1.8 to be 831.9 ng/mL (equiv. 6.0 nM/L), compared with that of 23.2 ng/mL (15.4 nM/L) for the smaller molecule $Pam_3CSK_4$ (Fig. 4i). Further, we asked whether acGM-1.8 could synergize or antagonize with $Pam_3CSK_4$. We stimulated the TLR2 reporter cells with $Pam_3CSK_4$ (1 to 100 ng/mL) alone or with $Pam_3CSK_4$ (1 to 100 ng/mL) together with acGM-1.8 (1 μg/mL). For $Pam_3CSK_4$, at each concentration point, its effect was enhanced by the co-existence of acGM-1.8; even if at the highest concentration, the effect of $Pam_3CSK_4$ did not overshadow that of acGM-1.8, suggesting that the two compounds might form a synergy in action (Fig. 4j). Meanwhile, pre-treatment of these cells with acGM-1.8 (1 μg/mL; 30 min) had no significant influence on the effect of $Pam_3CSK_4$, suggesting that the former did not antagonize with the latter (Supplementary Fig. 7c).

Moreover, because when TLR2 is activated, it forms dimeric complexes with either TLR1 or TLR6, we asked whether its activation by acGM-1.8 involved TLR1 or TLR6. We pre-incubated TLR2 reporter cells with the antibody of TLR1 or TLR6 for 24 h before adding acGM-1.8. Anti-TLR1 mildly (~20%) inhibited the activation of the cells by acGM-1.8, but the inhibition did not change as the dose of antibody increased. In contrast, anti-TLR6 more strongly weakened the response of the reporter cells; its inhibitory ratio increased from 46.6%, 53.7% to 62.7% as its dose increased from 100, 500, to 1000 ng/mL (Fig. 4k). Further, we found that pre-treatment of BMDMs with both anti-TLR2 and anti-TLR6 almost completely abolished the effect of acGM-1.8 ($Pam_2CSK_4$ was used as the TLR2/6 ligand, Supplementary Fig. 7d). These data indicated TLR6 to be the major co-receptor of TLR2 upon acGM-1.8 activation.

Finally, we asked whether the activity of acGM-1.8 is associated with endocytosis. After being co-cultured with FITC-labeled acGM-0.2 or acGM-1.8 (Supplementary Fig. 8a), BMDMs rapidly internalized the particle-shape acGM-1.8 but not the linear acGM-0.2 (57.6 vs 5.8%), and this internalization could effectively be blocked by MDC, a clathrin inhibitor (57.6 vs. 24.5%, Supplementary Fig. 8b and c). However, the blocking of endocytosis did not compromise the activity of acGM-1.8 in stimulating macrophages towards M1 polarization (Supplementary Fig. 8d), which implies that endocytosis might not play a key role in the acGM-1.8's function.

Together, these results suggest that acGM-1.8 specifically activates TLR2/6 signaling to stimulate macrophages towards M1 polarization.

**acGM-1.8 exhibits anti-tumor potential in mice.** We set out to examine the anti-tumor potential of acGM-1.8 in tumor-bearing mice, with the treatment procedure illustrated in Fig. 5a. Two types of tumors, S180 sarcoma and B16 melanoma, were subcutaneously inoculated in mice. When the tumor diameter reached 0.5 cm, the mice were randomly divided into four groups. PBS (Group i) or acGM-1.8 (10 mg/kg; Group ii, iii, and iv) was intratumorally injected every two days. A mouse with tumor exceeding 1.5 cm in size was judged to be dead and euthanized by strictly following the ethical guidance for experimental animals.

Our data demonstrated that acGM-1.8 effectively suppressed the growth of both S180 (Fig. 5b–e) and B16 (Fig. 5f–i) tumors in vivo. At day 14, all the tumor-bearing mice in Group i (PBS-treated) had died, but all those receiving acGM-1.8 treatment maintained alive (Group ii, iii and iv). After day 14, continued treatment with acGM-1.8 was both effective and necessary. The mice in Group iii, which had received acGM-1.8 by day 14 and were then switched to receiving PBS, started to die at day 19 and none survived by day 26; in contrast, all the mice in Group iv, which kept receiving acGM-1.8 injection, survived through the 28-day observation (S180: Fig. 5b; B16: Fig. 5f).

Gross view of the collected tumor samples (S180: Fig. 5c; B16: Fig. 5g) further verified the anti-tumor potency of acGM-1.8, which could markedly reduce the tumor size by day 14 (comparing Group i and ii). The data also confirmed the necessity to continue the treatment after day 14. The tumors re-developed after the termination of acGM-1.8 injection (Group iii) but were further controlled by the continued administration of acGM-1.8 (Group iv); in the latter, two out of five samples were eliminated.

Measurement of the tumor size (S180: Fig. 5d; B16: Fig. 5h) and weight (S180: Fig. 5e; B16: Fig. 5i) provided consistent findings. For instance, in the S180 group, at day 14, the tumors from acGM-1.8-treated mice were nearly half in mass of those from the control group (1.4 vs. 3.2 g); while at day 28, the samples collected from Group iv were below 0.4 g. Also, histological staining revealed large necrotic areas in the tumor samples from the groups treated with acGM-1.8 but not those treated with PBS (S180: Supplementary Fig. 9a; B16: Supplementary Fig. 9b).

Two additional tests were performed. One experiment examined the effect of acGM-0.2 in vivo by using the same protocol, and the results suggested that it had no anti-tumor effect (S180: Supplementary Fig. 9c–f; B16: Supplementary Fig. 9g–j). The other experiment assessed the influence of acGM-1.8-challenged macrophages on the viability of tumor cells in vitro. After treating BMDMs with acGM-1.8, acGM-0.2, or PBS for 24 h, we transferred the culture medium to pre-seeded S180 sarcoma or B16 melanoma cells and incubated for 48 h. Cell viability assay showed that acGM-1.8 reduced the viability of S180 and B16 to 60.7% (Supplementary Fig. 10a) and 67.5% (Supplementary Fig. 10b), respectively, while acGM-1.8 itself did not kill tumor cells (Supplementary Fig. 10c and d).

In summary, these data demonstrate that acGM-1.8, when intratumorally injected, could effectively suppress the growth of two tumor models in mice. According to our hypothesis, acGM-1.8 stimulates macrophages and thereby induces anti-tumor immune responses. Hence, we continued to examine whether acGM-1.8 could activate both innate and adaptive responses against the tumor.

**acGM-1.8 exerts anti-tumor activity through macrophage-mediated immune responses.** We investigated the changes in the immune context of the tumor niche in several aspects. First, we dissected the profiles of different immune cell populations in the S180-bearing mice after acGM-1.8 treatment. We observed an overall increase of leukocytes in the tumor ($CD45^+$; Fig. 6a). Among the different populations, the proportion of M1-type macrophages ($F4/80^+/CD11c^+$) increased from 26.3 to 34.5% in the acGM-1.8-treated sample (Fig. 6b), while that of M2-type macrophages ($F4/80^+/CD206^+$) decreased from 16.9 to 12.1% (Fig. 6c). This trend in macrophage polarization was desirable and consistent with the in vitro data on the macrophage-stimulating effect of acGM-1.8 (Figs. 2–4). The overall proportion of T cells in the tumor had no significant increase ($CD3^+$, Supplementary Fig. 11a); however, encouragingly, the number of both $CD4^+$ (Fig. 6d) and $CD8^+$ (Fig. 6e) T cells increased, while that of Treg population ($CD4^+Foxp3^+$, Fig. 6f) decreased, reflecting an activation of the adaptive immune response. Meanwhile, the content of $Ly6G^+$ cells (Supplementary Fig. 11b) increased, indicating an influx of neutrophils. Importantly, the percentage of both $CD4^+$ (13.8 to 23.0%; Supplementary Fig. 11c) and $CD8^+$ T (9.7 to 17.8%; Supplementary Fig. 11d) cells in the blood doubled after acGM-1.8 treatment, providing further evidence of the restoration of anti-cancer adaptive immunity.

Meanwhile, we analyzed the key cytokines in the tumor niche. The ELISA quantification data indicated a down-regulation of

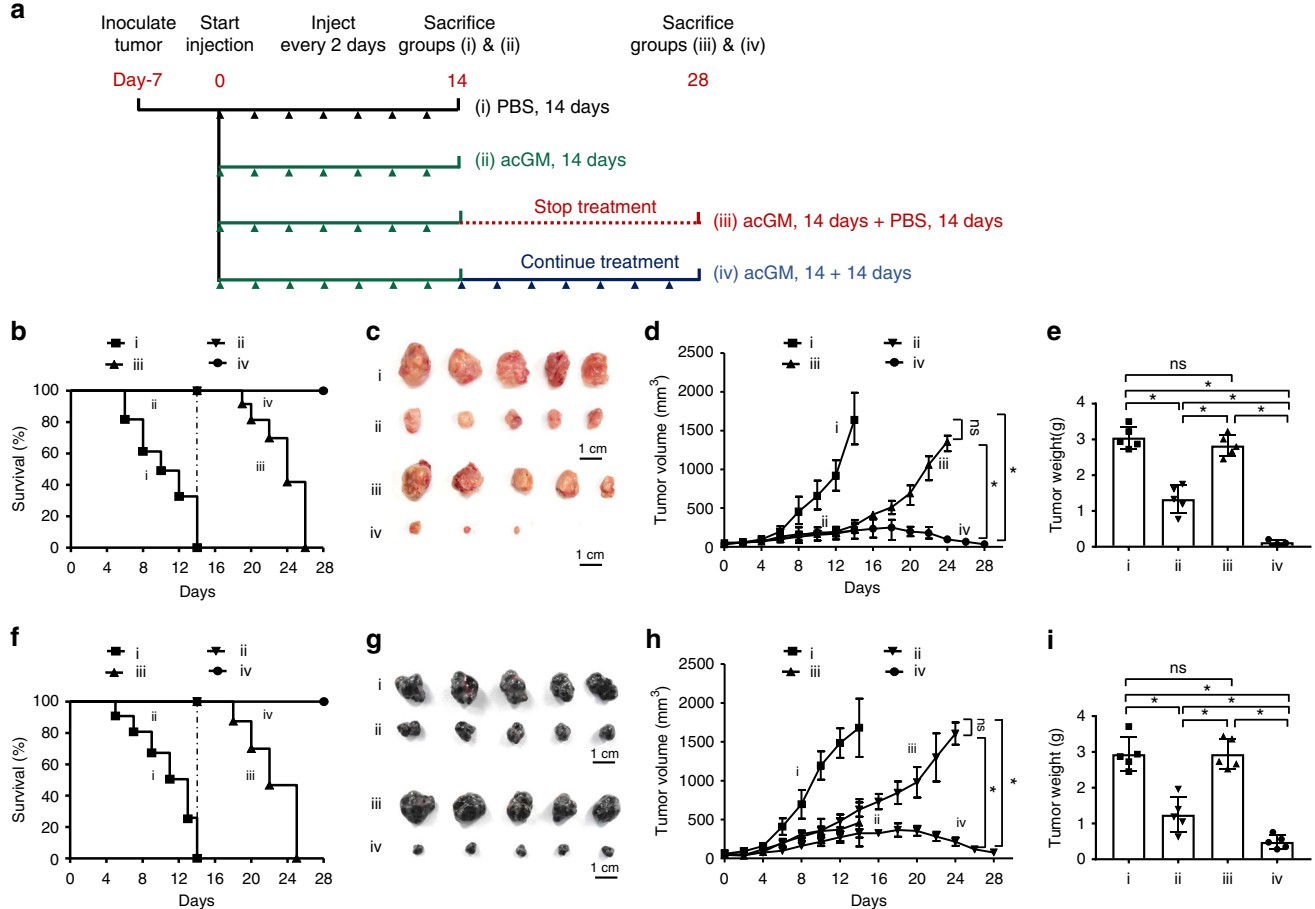

**Fig. 5** acGM-1.8 suppress the growth of two tumor models in mice. **a** Schematic illustration of the treatment procedure of acGM-1.8 in two types of tumors, S180 sarcoma and B16 melanoma, growing in mice. When the tumor diameter reached 0.5 cm, the mice were randomly divided into four groups and treated with: i) PBS for 14 days; ii) acGM-1.8 (10 mg/kg) for 14 days; iii) acGM-1.8 (10 mg/kg) for 14 days and then PBS for 14 days; iv) acGM-1.8 for 28 days. PBS or acGM-1.8 was intratumorally injected every two days. **b** Survival ratio of the S180 sarcoma-bearing mice in the four groups. **c** Gross view of the S180 tumor samples. **d** Measurement of the S180 tumor size. **e** Measurement of the S180 tumor weight. **f** Survival ratio of the B16 melanoma-bearing mice in the four groups. **g** Gross view of the B16 tumor samples. **h** Measurement of the B16 melanoma tumor size. (**i**) Measurement of the B16 melanoma tumor weight; *$P < 0.05$ between the two compared groups; ns: no significance; $n = 5$

pro-tumor cytokines (IL-10, VEGF-A, and TGF-β1) and an up-regulation of anti-tumor cytokines (TNF-α, IL-12 p70, and IFN-γ) in the acGM-1.8-treated group (Fig. 6g). Further, IF staining confirmed that acGM-1.8 treatment suppressed the secretion of IL-10 and VEGF-A and stimulated that of IFN-γ in the tumor tissue (Supplementary Fig. 11 e and f). IL-10, mainly produced by anti-inflammatory monocytes/macrophages, and TGF-β1 play key roles in establishing tumor immunosuppression; while VEGF-A, of which macrophages are also primary producers, orchestrates tumor angiogenesis. Among the anti-tumor cytokines, TNF-α and IL-12 p70 are chiefly secreted by macrophages – the former has a direct, strong tumoricidal effect and the latter is key to activating anti-tumor T cells; while IFN-γ, abundantly produced by CD4 (Th1) and CD8 cytotoxic T lymphocytes (CTLs), plays an essential role in establishing tumor immuno-surveillance. These data provide evidence that acGM-1.8 switches macrophages into an anti-tumor phenotype and consequently activates adaptive immune responses.

Then, we evaluated the roles of macrophages and T cells in this action. We first employed clodronate liposomes to deplete macrophages in situ in S180 tumor-bearing mice (Supplementary Fig. 12a). In these mice, the therapeutic activity of acGM-1.8 was abolished; no significant difference was observed in either the

tumor size (Fig. 6h) or tumor weight (Fig. 6i) between the groups treated by PBS and acGM-1.8, suggesting the crucial role of macrophages in the function of acGM-1.8. Then, we repeated the experiment in nude mice, where functioning T cells are absent, and found that acGM-1.8 could still reduce tumor size (Fig. 6j) and weight (Fig. 6k). However, the potency of acGM-1.8 was heavily weakened compared to that observed in normal mice (Fig. 5); the tumors kept growing, despite slower, in the treatment group. Further analysis confirmed that, in nude mice, acGM-1.8 could as well polarize the macrophages in the tumor from an anti-inflammatory to a proinflammatory phenotype (Fig. 6l and Supplementary Fig. 12b). These findings indicate that, for the therapeutic effect of acGM-1.8, macrophages are indispensable – they are the main cellular target and mediate the restoration of anti-tumor immune response; meanwhile, T cells also played a significant role in this action – in agreement with the outcomes from T cell profiling and IFN-γ determination.

Further, to validate that acGM-1.8 directly modulated the phenotype of the macrophages in the tumor, we examined the presence of Ly6C+ cells, together with that of M1 (CD11c+) or M2 (CD206+) cells, in the tumor tissue. Immunofluorescent staining revealed that the acGM-1.8 treatment markedly increased the number of M1 cells, but most of these M1 cells

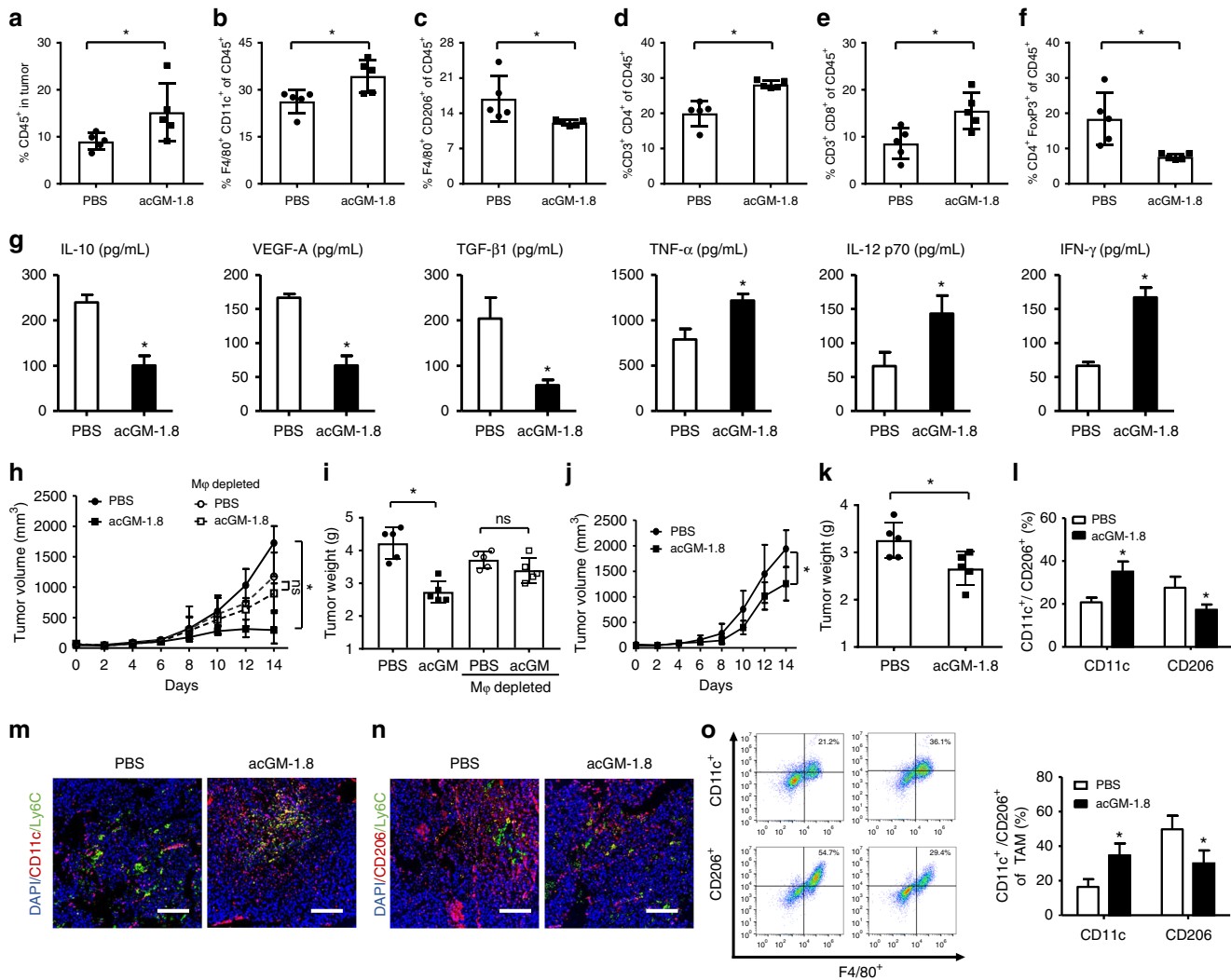

**Fig. 6** Macrophages mediate acGM-1.8's anti-tumor activity. **a–f** Profiling of immune cell populations in the S180 tumor of in mice. After the 14-day treatment by acGM-1.8 or PBS, the tumor tissue was harvested, processed, and analyzed with flow cytometry for the proportion of (**a**) leukocytes (CD45⁺) in tumor cells, (**b**) M1-type macrophages (F4/80⁺ CD11c⁺) in CD45⁺ cells, (**c**) M2-type macrophages (F4/80⁺ CD206⁺) in CD45⁺ cells, (**d**) CD4⁺ T lymphocytes in CD45⁺ cells, (**e**) CD8⁺ T lymphocyte in CD45⁺ cells, and (**f**) regulatory T cells (CD4⁺ Foxp3⁺) in CD45⁺ cells. **g** Determination of pro-tumor cytokines (IL-10, VEGF-A, and TGF-β1) and anti-tumor cytokines (TNF-α, IL-12 p70, and IFN-γ) by ELISA. **h, i** Evaluation of the anti-tumor effect of acGM-1.8 in S180-bearing mice with macrophages depleted by clodronate liposomes: measurement of (**h**) the tumor size and (**i**) weight after the 14-day treatment; circles and squares denote the PBS and acGM-1.8 treatment, respectively; solid and hollow signs represent mice without and with macrophage depletion, respectively. **j–l** Evaluation of the anti-tumor effect of acGM-1.8 in S180-bearing nude mice: measurement of (**j**) the tumor size and (**k**) weight after the 14-day treatment; *P < 0.05 versus the PBS treatment; n = 5; and (**l**) the proportion of M1/M2-type macrophages within CD45⁺ cells in the tumor tissue in the nude mice. **m, n** Representative images for co-staining of Ly6c⁺ (green) and (**m**) CD11c⁺ (red) or (**n**) CD206⁺ (red) cells in the tumor tissue of the S180-bearing mice; the cell nuclei were counter-stained with DAPI (blue); scale bar: 100 μm. **o** The effect of acGM-1.8 on the phenotype change of tumor-associated macrophages (TAM) ex vivo: macrophages were isolated from the tumor, cultured in vitro, treated with acGM-1.8 for 48 h, and analyzed for (**o**) their M1/M2 phenotype change by the flow cytometry; *P < 0.05 vs. the PBS treatment; ns: no significance; n = 5

did not co-express Ly6C (Fig. 6m and S13a); the treatment also decreased the amount of M2 population, and there was little overlap between the signals of CD206 and Ly6C (Fig. 6n and S13b). Ly6C⁺ cells were found in both acGM- and PBS-treated tumors, presented in a similar level (~ 15%). This group of data suggest that, though the Ly6C⁺ inflammatory monocytes can infiltrate into the tumor in both sample and control groups, these cells are unlikely to be the main target of acGM-1.8.

To further confirm that acGM-1.8 could directly convert tumor macrophages into M1 cells, we isolated macrophages from the tumors, cultured them ex vivo, and treated the cells with acGM-1.8 in vitro. Phenotype analysis showed that acGM-1.8 could directly increase the proportion of CD11c⁺ cells and decrease

that of CD206⁺ cells in these ex vivo cultured tumor macrophages (Fig. 6o). These data are consistent with the above findings and further suggest that acGM-1.8 modulated tumor macrophages more than infiltrating monocytes.

**acGM-1.8 is a TLR agonist with high safety for in vivo use.** Finally, we evaluated the safety of acGM-1.8 for in vivo use. We compared acGM-1.8 with the four classical molecules of its kind – LPS, MPLA (agonist of TLR4), Poly (I:C)(TLR3) and Pam₃CSK₄ (TLR1/2). We intraperitoneally injected them into mice at three doses and monitored for 24 h. Encouragingly, when applied at 20 mg/kg, acGM-1.8 exhibited its high safety as 9 out of 10 mice survived; in sharp contrast, only 1 out of 10 injected with

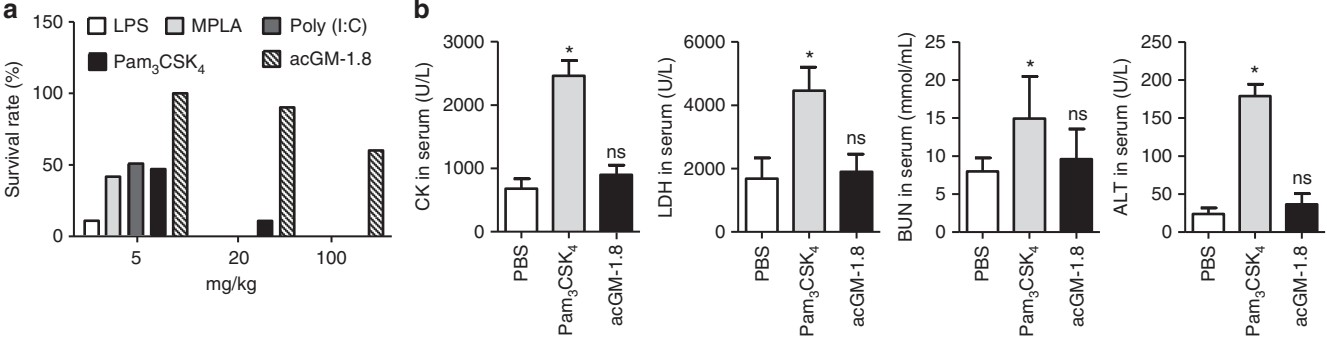

**Fig. 7** Assessment of the safety of acGM-1.8 in vivo. **a** acGM-1.8, or four classical TLR ligands: LPS, MPLA, Poly (I:C) and Pam$_3$CSK$_4$, was intraperitoneally injected to mice at the same dose. The survival rate was calculated after 24 h of injection; $n = 10$. **b** Measurement of the key biochemical parameters, including creatine kinase (CK), lactic dehydrogenase (LDH), blood urea nitrogen (BUN), and alanine transaminase (ALT), in the serum of the mice alive after injected with PBS, Pam$_3$CSK$_4$, or acGM-1.8 (20 mg/Kg); *$P < 0.05$ versus the PBS treatment; $n = 5$

Pam$_3$CSK$_4$ was alive, while administration of LPS, MPLA and Poly (I:C) at the same dose killed all the animals. When given at an extremely high dose of 100 mg/kg, the four other TLR agonists killed all animals, while still 6 were alive in the acGM-1.8 cohort (Fig. 7a). Measurement of the key biochemical parameters of the mice treated with Pam$_3$CSK$_4$ or acGM-1.8 (20 mg/kg) revealed that the levels of creatine kinase (CK), lactic dehydrogenase (LDH), blood urea nitrogen (BUN), and alanine transaminase (ALT) in the acGM-1.8-treated mice were similar with those in the control group (Fig. 7b). Finally, we observed the mice receiving acGM-1.8 at three doses for five days; our findings confirmed that acGM-1.8 administered at 5 and 20 mg/kg had a 100 and 90% survival rate, respectively (Supplementary Fig. 14a). Measurement of the animal weight (Supplementary Fig. 14b) and monitoring of the animal well-being (including daily activation and grooming) showed no obvious abnormality. Such findings underline the safety of acGM-1.8 for in vivo use, which can be a significant advantage over the conventional TLR2 or TLR4 agonists.

## Discussion

Inspired by the immunostimulatory microbial signals, in this study, we have designed and evaluated a novel PAMP-mimicking regent that can activate macrophage-mediated immunotherapy. Our results demonstrated that acGM-1.8, a glucomannan polysaccharide modified with acetyl groups with a substitution degree of 1.8, could stimulate macrophages to release proinflammatory cytokines. We further identified acGM-1.8 as a specific agonist of TLR2 and validated its efficacy in producing macrophages with anti-tumor potential in vitro and in vivo. Encouragingly, acGM-1.8 showed a higher safety than the established TLR2 agonists when tested in mice.

Many strategies in cancer immunotherapy are designed to target a broad variety of cells, receptors, and signaling pathways across the innate and adaptive immunity. Traditional opinions put the antigen-specific cancer-killing response from the adaptive immunity in the center of the stage and regard the innate immunity to exert limited roles of releasing cytokines to activate adaptive immunity[28,29]. These concepts are constantly evolving because of increasing new findings. Clinical evidence suggests that numerous types of cancers do not express distinct tumor antigens; in these scenarios, both the promotion and suppression of the tumor growth are regulated by innate immune mechanisms. Harnessing macrophage responses is being considered as a more powerful approach than before for effective immunotherapy[4,30]. Most recently, researchers reported the anti-tumor efficacy of a nanosystem delivering TLR7/8 agonist to target TAMs[31], as well

as successful reactivation of cytotoxic T cells to eliminate cancer cells by upregulating IL-12 p40 in macrophages[32,33]. In this study, we devised a TLR2 agonist with high specificity to stimulate macrophages into a phenotype that can – both i) exhibit direct killing of cancer cells by producing relevant cytokines (the traditional innate way of response) and ii) reactivate the T cell-mediated surveillance (the adaptive regulation). Our data showed that, through specific activation of macrophages, acGM-1.8 could restore a strong immune attack against the tumor models in vivo.

TLRs on macrophages (and other innate immune cells) play essential roles in immune regulation[34]. Various TLR ligands/agonists have been developed for therapeutic purposes such as cancer treatment[35]. Although some of them have been used in clinical trials[36], important issues remained with their efficacy and safety. First, the specificity of these microbial derivatives is low. MPLA, a chemically modified derivative of *S. minnesota* endotoxin, and Bacillus Calmette-Guérin (BCG), an extraction from *Mycobacterium tuberculosis*, activate both TLR2 and TLR4 – like LPS. For LPS, even the type of microbes it was derived from can affect its specificity[12]. Also, as both literature and our data showed, these established TLR agonists exerted high toxicity in vivo. Possibly as a consequence of the above two issues, the effect of a TLR agonist on tumor growth is inconsistent between studies, with unsatisfactory outcomes reported[37,38]. Although the exact reasons remain unclear, we suspected that the microbial origin and complex structural features of these molecules led to unspecific activation of the cell receptors and consequently unwanted expression of cytokines. Accordingly, we for the first time hypothesized and validated that a simple structure could be an ideal molecule – i) a macrophage-affinitive glycan pattern ('GM') modified with ii) the simplest aliphatic modification ('ac'). As a new TLR agonist, this molecule concisely recapitulates the essential features of PAMP, specifically activating TLR2 and exhibiting lower toxicity than all the tested classical TLR agonists in mice.

Our findings in vivo highlight the pivotal role of macrophages in mediating the therapeutic activity of acGM-1.8, while also underscoring the substantial contribution from T cells. The data from immunocytes profiling, cytokine determination and IF staining demonstrate that acGM-1.8 can directly switch the phenotypes of tumor macrophages, and the outcomes from the macrophage depletion assay offer key evidence that these cells are indispensable in acGM-1.8's function. Our finding is consistent with other recent opinions that highlight the central role of macrophages in orchestrating anti-tumor immune responses[37,39,40]. Intriguingly, the depletion of macrophages *per se* slightly slowed down the tumor growth, which might be a

consequence of the removal of a certain portion of pro-tumor macrophages. However, the inhibition resulted from macrophage removal was neither potent nor controllable, and the tumor size kept increasing if there was no further treatment; hence, macrophage depletion itself had no therapeutic implication in our model.

In addition to innate immune cells, T lymphocytes considerably contribute to the therapeutic effect of acGM-1.8. The increase in the numbers of CD4[+] and CD8[+] T cells in both the tumor niche and the circulation, together with the elevated amount of IFN-γ, marks the establishment of the adaptive immune response against the tumor[41]. This finding adds weight to the understanding that macrophages can initiate the cooperation between the innate and adaptive immune responses against the tumor; for instance, tumor macrophages switching from M2 to M1 phenotype can activate Th1 response and induce tumor rejection[4,42]. The weakened potency (though still considered effective) of acGM-1.8 observed in the nude mice highlights the contribution of T cells. Together, these data suggest that acGM-1.8 unleashes the anti-tumor potentials of macrophages that consequently restore the T cell-mediated anti-tumor responses in vivo.

On the basis of the encouraging efficacy, specificity and safety of acGM-1.8, future studies can be directed in several aspects. First, macrophages are highly plastic. Although acGM-1.8 can trigger these cells to express the desirable group of proinflammatory factors through activating TLR2, it should be noted that the TLR2 activation precipitates multiple signaling pathways and intracellular events. Certain cytokines can be a double-edge sword depending on their dose of expression. Hence, more precise control of the duration and threshold of TLR activation is needed. Second, in this proof-of-concept study, we used intratumoral injection which is straightforward. However, in future trials, for tumors in different types and under different developmental stages, the means and dose of administration need to be tailor-made. Third, acGM-1.8 may be used in combination with other chemotherapeutic, immunotherapeutic or radiotherapeutic means, or can be further developed into adjuvants for more therapeutic opportunities.

## Methods

**Preparation of acetyl/deacetyl glucomannan.** KGM (1 g) was adequately swollen in water (200 mL) and freeze-dried. The mixture of anhydride and pyridine (1:1, V/V) 100 mL was stirred at 50 °C and slowly poured into round-bottom flask with KGM. After specific time (6 to 72 h) of reaction, distilled water (10 mL) was added to stop the reaction, and hydrochloric acid (HCl) was used to neutralize the solution. The products were precipitated overnight with ethanol and filtrated, repetitively re-suspended with ethanol, centrifuged again for 5 times, and lyophilized to obtain the final product of acetyl GM (acGM). Acetyl dextran (acDex, Mw 100,000) was synthesized using the same method. To obtain deacetyl GM (deGM), DBU (0.5 mL) was added to a solution of KGM (0.2 g) in a mixture of DMF and methanol (4:1) at 40 °C and stirred for 2 h. All products were characterized by NMR and IR spectrum[43].

**Preparation of alkylated glucomannan.** KGM (1 g) was dissolved in water (200 mL) and freeze-dried. A pre-mixed solution of TFAA and alkyl acid (butyryl, hexanoyl, octanoyl, and decanoyl acid) was stirred at 50 °C for 20 min before freeze-dried KGM was immediately added to the flask. The mixture was stirred at 65 °C for 2 h under nitrogen and precipitated by ethanol. The product was filtered, repetitively dissolved in chloroform (CHCl₃), and precipitated in ethanol for 5 times before vacuum drying. All products were characterized by NMR spectrum[44].

**Formation of assembled glucomannan in water.** A stock solution of acGM or alkylated GM (acGM were dissolved in dimethylsulfoxide, alkylated GM were dissolved in CHCl₃, 2 mL, 4 mg/mL) were prepared, and dropwise into water (10 mL) under sonicating for 2 min on ice by using a probe sonicator (T-10, IKA, Germany), and magnetic stirred overnight for stabilization, then dialysed against deionized water in dialysis tubes (Mw 3,500) until no dimethylsulfoxide (DMSO) was detected. Polysaccharide particles were obtained by lyophilisation and resuspended in water and the concentration was adjusted to 2 mg/mL. Different size of acGM-1.8 (200 nm to 1200 nm) were prepared by adjusting the concentrations (in DMSO) of stock solutions.

Particle sizes, distributions and ξ-potential were measured by dynamic light scattering using Zetasizer Nano ZS (Malvern Instruments, United Kingdom) with three replicates. The morphology of the nanoparticles was observed by transmission electron microscopy (TEM, JEOL Ltd., Tokyo, JAPAN).

**Preparation of acetyl dextran.** Dextran (Dex) was swollen in water (200 mL) and freeze-dried. The mixture of anhydride and pyridine (1:1, V/V) 100 mL was stirred at 50 °C and slowly poured into round-bottom flask with Dex. After specific 48 h of reaction, distilled water (10 mL) was added to stop the reaction, and hydrochloric acid (HCl) was used to neutralize the solution. The products were precipitated overnight with ethanol and filtrated, repetitively re-suspended with ethanol, centrifuged again for 5 times, and lyophilized to obtain the final product[44].

To obtain acDex-1.9 particles, acDex-1.9 was dissolved in DMSO (2 mL, 4 mg/mL) and dropwise into water (10 mL) under sonicating for 2 min on ice by sonicator and magnetic stirred overnight, then dialysed against deionized water in dialysis tubes (MWCO 3,500 Da) until no DMSO was detected. AcDex-1.9 particles were obtained by lyophilization and resuspended in water and the concentration was adjusted to 2 mg/mL. Particle sizes, distributions and ξ-potential were measured by dynamic light scattering using Zetasizer Nano ZS (Malvern Instruments, United Kingdom) with three replicates.

**Determination of degree of substitution (DS).** The DS of acGM was determined by using a hydroxylamineeferric trichloride method. One milligram sample was precisely weighed and added into the 50 mL volumetric flask, then, hydroxylamine hydrochloride (0.1 M, 5 mL) and NaOH (1.5 M, 5 mL) was added and reacted for 20 min, HCl (2 M, 5 mL) was subsequently put in and stood for another 10 min for neutralizing. Finally, FeCl₃ (0.37 M, 10 mL) was added and diluted with water to 50 mL. The absorbance was measured at 500 nm. While different concentration of β-D-acylated glucose solution was used to define standard curve.

The degree of substitution of acetyl groups could be calculated from the acetyl content as below, and each group had three replicates:

$$M = [mAc/(mAc + 1000)] \times 100\%$$

$$DS = 162M/(4300 - 42M)$$

mAc: the amount of acetyl group per milligram (μg).
M: the content of acetyl group of acGM (%).
DS: the average number of acetyl substituent attached per sugar unit of GM.

**Cell lines.** Mouse sarcoma cell line (S180) and melanoma cell line (B16-F10) were purchased from American Type Culture Collection (ATCC). HEK-Blue-mTLR4 (Cat No: hkb-mtlr4), HEK-Blue-mTLR2 (Cat No: hkb-mtlr2) and HEK-Blue-Null cells (Cat No: hkb-null2) were purchased from InvivoGen (France) and maintained according to the vendor's instructions.

**Animals.** Female C57BL/10 J mice (6–8 weeks old) and nude mice were purchased from the Model Animal Research Center of Nanjing University (China), TLR4[-/-] mice (Tlr4[lps-del], C57BL/10ScN background) and TLR2[-/-] mice (B6.129-Tlr2[tm1Kir/J], C57BL/6 J background) were purchased from Jackson Laboratory (USA) through Nanjing Biomedical Research Institute of Nanjing University (China). All animals were raised in specific-pathogen-free animal rooms and were treated according to the local policy for animal experiments. The animal protocols were reviewed and approved by the Animal Care and Use Committee of Nanjing University and University of Macau, respectively, and were conformed to the Guidelines for the Care and Use of Laboratory Animals published by the National Institutes of Health, USA.

**Primers and antibodies.** All primers used for RT-qPCR were synthesized by Life techologies (China), and their sequences were listed in Supplementary Table 1. The antibodies used in this study are listed in Supplementary Table 2.

**In vitro assessment of acGM on macrophages.** BMDM were used to evaluate the bioactivity of acGM. The primarily derived and characterized BMDM were either directly used or pre-induced into a M2 phenotype by treatment with IL-4 (40 ng/mL) and IL-13 (20 ng/mL) for 48 h. The cells were seeded on a six-well culture plate (each well contained 2 × 10⁶ cells) and stimulated with GM samples (100 μg/mL) for 24 h. Then, the culture medium was collected, and the levels of various secreted cytokines were determined by enzyme-linked immunosorbent assay (ELISA), according to the manufacturer's instructions (Neobioscience Technology, China); in parallel, the cells were gently rinsed with cold PBS and their RNA was collected with TRIzol for subsequent RT-qPCR analyses. The whole gene expression (microarray) was assessed in RayBiotech, Inc. (Guangzhou, China) with Agilent Whole Mouse Genome Oligo Microarray Kit. Pathways related to inflammation were listed and heat map was generated to analyze the different

stimulations between acGM-0.2 and acGM-1.8 in toll-like receptor signaling pathway.

At the same time, BMDM without further polarization (BMDM-M0) or those induced into M2 phenotype (BMDM-M2) were seeded on 6-well culture plate ($2 \times 10^6$ cells per well) and treated with acGM-1.8 of increasing concentrations (20, 50, and 100 μg/mL); after 48 h, cells were collected with a scraper and analyzed by flow cytometry for F4/80, CD11c, CD206, and CD163.

**Examination of TLR signaling pathway.** The activation of TLR signaling was evaluated by quantifying the level of secreted embryonic alkaline phosphatase (SEAP) produced by two NF-κB reporter cell lines (HEK-Blue-mTLR2 cells and HEK-Blue-mTLR4 cells), with their parental cell lines (HEK-Blue-null1-v and HEK-Blue-null2 cells) as negative control. acGM were incubated with the reporter cells for 24 h and the SEAP levels were quantified by incubating supernatant with Quanti-Blue substrate for 12 h and read at 650 nm according to the manufacturer's protocol. LPS (20 ng/mL) and Pam$_3$CSK$_4$ (100 ng/mL) were employed as positive control.

Peritoneal macrophages harvested from wild type mice, TLR-2 KO mice, or TLR-4 KO mice were seeded on 6-well culture plates ($2 \times 10^6$ cells per well) and incubated with acGM (DS = 1.8) for 24 h. The related cytokines in supernatant were determined by ELISA.

**Determination of EC$_{50}$ of acGM-1.8.** To determine the 50% effective concentration (EC$_{50}$) value of acGM-1.8, 20 μL of Pam$_3$CSK$_4$ (0 to 100 ng/mL) and acGM-1.8 (100 to 2000 ng/mL) were added into 180 μL of HEK-Blue-mTLR2 cells ($5 \times 10^4$ cells per well) seeded in a 96-wells plate. After a 24-h incubation, the SEAP levels were quantified according to the manufacturer's protocol. The EC$_{50}$ value was calculated by GraphPad Prism software. The model is Y = bottom + (top-bottom)/(1 + $10^{x\text{-Log IC50}}$). Where $x$ is the log concentration of samples, bottom is the lowest absorbance, and top is the maximum absorbance.

**Evaluation of competition between acGM-1.8 and Pam$_3$CSK$_4$.** HEK-Blue-mTLR2 cells ($5 \times 10^4$ cells per well) were stimulated with increasing concentrations of Pam$_3$CSK$_4$ alone or Pam$_3$CSK$_4$ along with acGM-1.8 (1 μg/mL) for 18 h; alternatively, cells were pre-treated with acGM-1.8 (1 μg/mL) for 30 min followed by stimulation with increasing concentrations of Pam$_3$CSK$_4$ for 18 h. The level of SEAP after these two treatments was quantified according to the manufacturer's protocol.

**Assessment of TLR1 and TLR6 as co-receptors.** TLR2 reporter cells were pre-incubated with anti-TLR1/TLR6 for 24 h (100, 500, or 1000 ng/mL). Then, acGM-1.8 (100 ug/mL) was added and incubated for another 12 h, before the culture medium was collected and measured. Inhibition rate (%) = [(A − B) / A] × 100%.

$A = a_{(OD650)} - c_{(OD650)}$, $B = b_{(OD650)} - c_{(OD650)}$,

where $a$ means the value of OD650 only with acGM-1.8, $b$ means the value of OD650 pre-incubated antibody before acGM-1.8, $c$ means the background of OD650 of culture medium.

BMDM were pre-treated with anti-TLR6 and anti-TLR2 (1000 ng/mL) for 24 h. Then, acGM-1.8 (100 μg/mL) or Pam$_2$CSK$_4$ (100 ng/mL; an agonist for TLR2/TLR6) was added for another 48 h. The cells were collected and analyzed by flow cytometry for CD11c and F4/80.

**Validation of the binding between TLR and acGM-1.8.** (1) Co-immunoprecipitation and detection for polysaccharides: MAb-mTLR2 (Invivogen), a mouse IgG1 against TLR2, was first mixed with Protein G PLUS-Agarose beads (Santa Cruz Biotechnology, Inc.) at 4 °C for 30 min. After acGM-1.8 or aCGM-0.2 (5 or 10 mg/mL) was mixed with the same amount of cell lysate on a vortex mixer at 4 °C for 1 h. The pre-treated beads were added and incubated at 4 °C on a rotating device overnight. Samples were boiled for 3 min and centrifuged to collect the supernatant. The obtained samples were subjected to sulfuric-phenol assay for detection of polysaccharides, as reflected by the absorbance measured at 490 nm. (2) Protein pull-down and Western blotting. HEK-Blue-mTLR2 cells were collected and washed twice with PBS before resuspension in ice-cold lysis buffer for 30 min. Then, the cell membrane protein was extracted using a specific extraction kit (Beyotime, China). The concentration of the protein was measured using BCA assay (R&D Systems, USA). Subsequently, acGM-1.8, −1.2, −0.6, or −0.2 (10 mg/mL) was mixed with the acquired proteins on a vortex mixer at 4 °C for 12 h before centrifugation ($10000 \times g$) to remove unbound proteins. The precipitates were washed with cold PBS for three times, and the proteins were eluted by the sample buffer and transferred to PVDF membranes (Bio-Rad, USA). After being blocked in bovine serum albumin (5%) for 1 h at room temperature with gentle shaking, the membrane was blotted with anti-TLR2 (1:1000, Cell Signaling, USA) at 4 °C overnight and incubated with secondary antibody (1:2500) for 2 h at room temperature. The bands were visualized with the SuperSignal West Pico Chemiluminescent Substrate (Thermos Scientific, USA)[45].

**In vivo antitumor assessment of acGM-1.8.** The S180 sarcoma and B16 melanoma cells were subcutaneously inoculated ($6 \times 10^5$ cells per mouse) under the left

armpits of mice. When the tumor diameter reached 0.5 cm, the mice (40 in total) were randomly divided into four groups (10 in each group) and injected intratumorally with: Group i) PBS (50 μL) for 14 days; ii) acGM-1.8 (5 mg/mL, 50 μL) for 14 days; iii) acGM-1.8 for 14 days and then PBS for another 14 days or until death; iv) acGM-1.8 for 28 days. Injection was performed every other day. The end of treatment for Group i) and ii) was set at day 14, while that for Group iii) and iv) was set at day 28. A mouse was judged "dead" when its tumor diameter exceeded 1.5 cm and euthanized, in strict accordance with the animal ethics guidance.

The tumor size was measured every two days with a caliper. At the points of termination, the tumor tissue from five mice in each group was collected, photographed and weighed. Approximately half of the samples were smashed. Then, a portion of each sample was incubated with collagenase IV (0.2% wt. in hanks' balanced salt solution [HBSS]) and desoxyribonuclease (0.1% wt.in HBSS) at 37 ºC for 45 min, treated with red blood cell lysis buffer (10 min) on ice, centrifuged, and filtered (70 μm filter). Then, the acquired single cells were divided into two parts; one part was subjected to flow cytometry analysis for the proportion of CD45$^+$ cells in the tumor, and the other part was applied to density gradient centrifugation to harvest leukocytes (mouse tumor infiltrating tissue leukocyte separation kit, TBD Science, China). The collected leukocytes were washed with cold PBS and stained with related antibody for flow cytometry analysis (See Supplementary Table 2 for antibody information).

Meanwhile, another portion of the smashed tumor sample, after adjustment to the same weight, was homogenized in PBS and centrifuged to collect the supernatant for the determination of related cytokines (IL-10, VEGF-A, TGF-β1, TNF-α, IL-12 p70, and IFN-ϒ) by ELISA.

Further, the other half of the collected tumor samples, without smashing, were paraffin-embedded and sectioned (6 μm) for histological (Hematoxylin and eosin, H&E) and immunofluorescent (IF) staining for IL-10, IFN-ϒ, VEGF-A, Ly6C, CD11c, and CD206.

Before the collection of tumors, peripheral blood was sampled from the mice by removing eyeballs for analyzing the proportion of CD4$^+$ and CD8$^+$ T lymphocytes by Flow cytometry.

In another set of experiments, acGM-0.2 (5 mg/mL, 50 μL) was also tested in the same way as acGM-1.8 for 14 days.

In a further set of experiments, nude mice were employed to assess the role of T cells in the anti-tumor functions of acGM-1.8. All the related treatment and sample collection followed the protocols described above.

**Depletion of macrophages in S180 sarcoma-bearing mice.** When the tumor diameter reached 0.5 cm, clodronate liposomes (5 mg/mL in PBS, 50 μL) was intratumorally injected 1 day before the injection of acGM-1.8 or PBS. Successful depletion of macrophages was validated by digesting the tumor tissue and analyzing the proportions of CD11b$^+$ and F4/80$^+$ macrophages with flow cytometry. All the related treatment and sample collection followed the protocols described above.

**Isolation of tumor-associated macrophages ex vivo.** The tumor tissue from S180 sarcoma-bearing mice was digested and applied to gradient centrifugation to harvest mononuclear cells (Mouse tumor infiltrating tissue mononuclear cell separation solution kit, TBDscience, China). The collected cells were seeded on a 6-well culture plate (each well contained $2 \times 10^6$ cells) for 2 h, and the non-adherent cells were removed. PBS or acGM-1.8 (100 μg/mL) was added to the culture medium and incubated for another 24 h. Then, the cells were scraped off and collected and analyzed for CD11c, F4/80, and CD206 with flow cytometry.

**In vivo safety assessment of acGM-1.8.** To analysis the safety of acGM-1.8, different TLR ligands including LPS, Pam$_3$CSK$_4$, MPLA, and acGM-1.8 were intraperitoneally injected to mice (10 mice in each group) in different doses (5, 20, and 100 mg/kg). After 24 h, the survival rate was measured. The mice weight was monitored for 5 days. The untreated and acGM-1.8-treated (20 mg/kg) mice were sacrificed. The serum from PBS, Pam$_3$CSK$_4$, and acGM-1.8 groups (20 mg/kg) were collected and the levels of CK, LDH, BUN, and ALT were measured by corresponding kits.

**Statistics.** Data are shown as Mean ± standard error of the mean (s.e.m.). All data were normally distributed. Statistical analysis was performed using Prism Software (GraphPad, USA). Student's t-test were performed, expect that two-way ANOVA was performed in Fig. 4j, Fig. 5d, h, Fig. 6h and Fig. 6j followed by Bonferroni's multiple comparison test. Results were considered significant at *$p < 0.05$.

**Reporting summary.** Further information on experimental design is available in the Nature Research Reporting Summary linked to this article.

## Data availability

The data of this study are available from the corresponding authors upon reasonable request. The microarray data for Fig. 4a–c and Supplementary Fig. 6a are available on Gene Expression Omnibus (GEO), GSE129591. The source data underlying Figs. 2b–g, 3c–e and h, 4d–k, 5 b, d–f, h and i, 6a–l and o, Fig. 7, and Supplementary Figures 2, 4, 6b

and c, 7b and d, 8, 9, 10, 13 and 14 are deposited in the Figshare database (https://figshare.com/s/e12bd5b0e0727024e152).

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

## Acknowledgements

This study was financially supported by funding grants from the Science and Technology Development Fund, Macao SAR (FDCT 080/2016/A2, 126/2016/A3, and FDCT Operating Fund for State Key Laboratory of Quality Research in Chinese Medicine) and the University of Macau Research Committee (MYRG2016–00031-ICMS-QRCM and MYRG2017–00028-ICMS). L.D. acknowledges funding supports from the National Key Research and Development Program of China (Grant No. 2017YFC0909702), the National Natural Science Foundation of China (Grant No. 31671031), the Jiangsu Province Funds for Distinguished Young Scientists (Grant No. BK20170015), and the Fundamental Research Funds for the Central Universities (Grant No. 020814380088, 020814380115). We thank Prof Xin Chen, Prof Renhe Xu, Dr Enqin Li, and Ms Tianzhen He at the University of Macau for their technological help and Miss Lina Li for her design of illustration.

## Author contributions

C.W. and L.D. designed the study. Y.F. and R.M. performed in vitro and in vivo biological experiments. Y.F. and P.X. performed chemical experiments. Z.W. performed part of the in vivo tests including the nude mice assay. Y.F. and C.W. drafted the manuscript. J.Z., L.D. and C.W. provided funding supports. All authors contributed to data analysis and manuscript drafting.

## Additional information

**Competing interests:** The authors declare no competing interests.

