## [Peer Review File · Nature Communications]

Reviewers' Comments:

Reviewer #1:

Remarks to the Author:

In this article, Feng et al have generated a TLR-2 agonist called acGm that can suppress tumor growth and had high safety. The way to design this compound is creative and straightforward, and it appeared to be safer than conventional bacterial products. In general, this work will be of broad interest to the readers of nature communications, but the following issues should be addressed:

1. Macrophage phenotype changes were extensively studied in this work. However, in vitro cultured macrophages can be very sensitive, and their "initial" phenotype must be clear. The authors showed that acGm could induce the macrophages into a TNF/IL-12-producing phenotype, but what was the level of these cytokines before stimulation?
2. Also about the macrophage stimulation: the data in Fig. 1f were promising, showing the compound inducing the M1 polarization in M2 cells. However, the data in Fig. 2f showed M2 induction was not successful? Or did the authors mean that the M2 type of induction? Why was acDex showing different results in two datasets?
3. The data in Fig. 3j about TLR2 activation raise several questions, why was the effect different between the two acGm groups? Additionally, why was the x-axis in 10 and 100? How was the statistics performed in comparing the three samples?
4. Speaking of the specificity of TLR agonists, the reviewer believes that cross-activation always happens, which may be a consequence of evolution. Is there any other specific TLR2 agonist either from bacteria or fungi used for cancer research? The authors should add more discussion on this issue with new references.
5. Minor issues and suggestions
 - a) for safety tests (fig. 6), in the groups where mice died, did they die immediately upon injection and simultaneously? When the authors show survival rate, was it calculated at the same timepoint?
 - b) figure legends should be improved: fig. 2 g, more explanations should be added; fig. 3 b and c, do the colored circles have the same meaning? fig. 4 4 and f, the circles/triangles should be explained. fig. 5 a has the same issue
 - c) fig. 6 a can be drawn better. The "mg/kg" sign is confusing under the data column
 - d) the carbohydrate units of glucomannan should be described.

Reviewer #2:

Remarks to the Author:

The manuscript reported a safe toll-like receptor agonist mimicking microbial signal to generate tumor-killing macrophages. The authors demonstrated the synthesis of acGM with different acetylation degrees and the capacity to recognise and activate macrophage TLR2 receptor with optimized acetylation degrees and structure. The key factors of material synthesis was discussed with experimental evidences and the mechanism of the tumor associated macrophages polarization was also explored. The study was comprehensive. The conclusion was supported with systematic experimental data. The technique can be very useful. I suggest that the manuscript should be published after considering the following comments.

- 1) I would suggest that the term "tumor-killing" in the title and the entire manuscript should be revised to be "tumor-suppressive", for accuracy.

- 2) Figure 2g describes the acGM dispersion change with increased pH. What will happen with decreased pH? What is the pH range for stable acGM particles? What is the stability of these acGM particles in PBS? These will be associated with the future application of the technique, so it is good to investigate them.
- 3) Beyond DLS measurement, it is good to give a size analysis as well from TEM images.
- 4) The direct binding between acGM and TLR2 is attractive. The presented co-immunoprecipitation (co-IP) data are useful, but Isothermal titration calorimetry (ITC) is mostly used to test interactions between smaller molecules including proteins. More investigation can be done to demonstrate the TLR2 binding to acGM with different modification degrees.
- 5) In the cell biology part: did the authors see any changes of cell polarization after MDC treatment? There was probably a link between blocking endocytosis and phenotype change. How did the authors gate all CD4+/CD8+ in flow cytometry data?
- 6) The authors mainly studied acGM0.2 and acGM1.8. What about acGM3.0? It looks like acGM3.0 also self-assemble to uniform nanoparticles and give good results shown in Fig. 1. Please provide an explanation.
- 7) The authors should compare this acGM with other reported systems to strengthen the discussion. It is also good to know the authors' views of the continual development of using acGM for anticancer therapy.
- 8) Some figure legends are incomplete (e.g., the safety data).

Reviewer #3:

Remarks to the Author:

Changing the phenotype of tumor associated macrophages towards a tumoricidal/anti-tumor immunity sustaining phenotype is an important hurdle to be taken in the immunotherapy of cancer. The work of Feng et al., therefore, is addressing an important issue. They have designed a compound which based on a number of in vitro assays and in vivo safety tests has some desirable features and maybe one of the compounds that truly helps to reach this goal. There are several issues that need to be addressed for this study to really show that their compound works as indicated by the authors.

1. In the abstract it is claimed that this compound – when intratumorally injected – eliminates two different tumors in vivo. This is not true, growth retardation of the tumors is shown, up to 14 days after challenge. At this point, this conclusion is not sustained by the data.
2. There are two statements concerning the antitumor activity of the macrophages stimulated with this compound, however, neither was a strong antitumor effect shown in vitro (Supplemental data about 50% at best) and there is no evidence that these macrophages mediate an antitumor effect in vivo (see also below). Hence, also these statements are – at this point – not supported by the data.
3. Last sentence Introduction: With anti-tumor immunity one means the adaptive immune response, the induction of anti-tumor T cells or alleviation of their effector function (restoration) but this has not been shown.
4. Page 6, did the macrophages also express the well-known M2 marker CD163 next to CD06 and this they produce VEGF?
5. Figure 4: please also provide schematic drawing of the treatment procedure in this figure.
6. Figure 4, the exact treatment is not clear, where mice injected until a certain day or until dead? And so on, a better description is required.
7. Figure 4, several remarks: a) Also provide tumor growth curves, 2. what happens when treatment stops? 3. What happens in the end, do all tumors disappear with continuous treatment? A longer observation period would be very informative with respect to the use of this compound in vivo.
8. Figure 5: The immune analyses need to be strongly improved. The reader needs to have the following information: a) what is the increase in CD45 cells?; b) Is there besides the increase of M1 also an increase of M2 macrophages, Ly6G+ cells and eosinophils's?; c) do the M1

macrophages express Ly6c, hence are they inflammatory and attracted to the tumor or are they really converted TAMs?; d) What is % of T cells within the CD45+ cell population, this needs to be established, and also their phenotype.

9. Figure 5c: quantification of the cells in several areas of the tumor and of several mice is required.

10. Figure 5: ELISA data, was there also production of TNF α , IL-12 and/or TGF β ? It would be nice to have this as it would connect the data with the in vitro characterization provided.

11. Last sentence page 13: this conclusion can't be made as "acGM-1.8 can efficiently generate M1-type anti-tumor macrophages and subsequently restore the T cell-mediated adaptive one against the tumor in vivo" was not shown. To establish this the authors need to: a) establish that this is the work of antitumor macrophages, by depleting macrophages or better use a mouse in which M1 macrophages can't do their job anymore (e.g. iNOS ko mice).; b) Furthermore, to show that this is not the work of T cells, they need to do depletion experiments for T cells.

12. The discussion needs to be adapted after these essential data sets requested have been provided.

Minor:

- A careful check of the English grammar should be performed. Some words are not correct, for instance "dissembled" instead of "disassembled" and "cultural medium" instead of "culture medium". But also some sentences are not correct. This is also observed in Legends.
- The figure legends of main and supplemental figures do not describe the data presented in such a way that the figures can be understood without having to look at the main text. Please provide better legends.
- Page 9, line 5 instead of stating "cells" state "BMDMs" to make it more clear what you used.

POINT-TO-POINT RESPONSES TO REVIEWERS' COMMENTS

Manuscript No. NCOMMS-18-34327A

We highly appreciate the three experts for their valuable suggestions. In the past three months, we have added new data and substantially revised our manuscript.

We are pleased to share our responses below.

Reviewer #1: Expert in Cancer Pharmacology

In this article, Feng et al have generated a TLR-2 agonist called acGm that can suppress tumor growth and had high safety. The way to design this compound is creative and straightforward, and it appeared to be safer than conventional bacterial products. In general, this work will be of broad interest to the readers of nature communications, but the following issues should be addressed:

Q1. Macrophage phenotype changes were extensively studied in this work. However, in vitro cultured macrophages can be very sensitive, and their initial phenotype must be clear. The authors showed that acGm could induce the macrophages into a TNF/IL-12-producing phenotype, but what was the level of these cytokines before stimulation?

A1. We thank the reviewer for this question. We confirm that throughout the study we had a clear understanding of the initial phenotype of the macrophages.

- 1) As we stated in Supporting Information, our laboratories routinely check the key polarization markers of the isolated macrophages (considered 'M0' for their initial phenotype), before inducing them into M1 or M2 state by using RT-PCR and FACS. One such example can be illustrated below (Figure-for-response-letter-only 1).
- 2) For the experiments in this study, we provided appropriate control: as shown in Figure 1 b and c in the manuscript, the macrophages treated with PBS expressed a lower level of TNF- α and IL-12 and a higher level of TGF- β and IL-10 than those stimulated with acGM-1.8. Other supporting data can be found in Figure 1f and Figure S2.

Figure-for-response-letter-only 1. RT-PCR (left) and FACS (right) checking of the murine bone marrow-derived macrophages (TNF- α , IL-12 and CD11c as M1 & TGF- β , IL-10 and CD206 as M2 markers). We routinely perform this test in our laboratories.

Q2. Also about the macrophage stimulation: the data in Fig. 1f were promising, showing the

compound inducing the M1 polarization in M2 cells. However, the data in Fig. 2f showed M2 induction was not successful? Or did the authors mean that the M2 type of induction? Why was acDex showing different results in two datasets?

A2. We apologize for this confusion which was caused by our unclear labelling in Figure 2f. As the reviewer wisely pointed out, the data in Figure 1f showed effective polarization. However, in Figure 2f, the label 'M2' does not mean M2-type induction; instead, it aimed to mean the cells used in this group had been pre-polarized into an M2 phenotype before the acGM treatment. We can see that acGM successfully induced these cells to express higher M1 markers. Meanwhile, the control material acDex could not reverse these cells into an M1 phenotype – and this finding is consistent.

To avoid this confusion, we have changed the label 'M2' in the original Figure 2f into 'untreated' in the revised Figure 2f and added 'BMDM-M2' at the top of the columns. We have also done a similar change to Figure 2e to avoid any misunderstanding.

Q3. The data in Fig. 3j about TLR2 activation raise several questions, why was the effect different between the two acGM groups? Additionally, why was the x-axis in 10 and 100? How was the statistics performed in comparing the three samples?

A3. To answer the first question: we apologize that our previous presentation should have been clearer. Here, the 'two acGM' groups meant different things – one meant that the cells were treated with acGM (at one concentration) and Pam₃CSK₄ (at increasing doses), and the two agents were added together; while the other meant that the cells were pre-treated with acGM (at one concentration) and then Pam₃CSK₄ (at increasing doses). This is a common approach to evaluate whether a new agonist synergizes or antagonizes with a known agonist (e.g. *J Biol Chem*, 2010, 285, 23755).

To make it clearer, we split the original Figure 3j into two figures – Figure 3j and Figure S7c. As shown in the new Figure 3j, the activity of Pam₃CSK₄ (alone) increased when its concentration became higher. At each concentration point, the co-existence of acGM-1.8 (1 µg/mL) enhanced the effect of Pam₃CSK₄; and this enhancement kept existing as the concentration of Pam₃CSK₄ rose. The data indicated that the effect of Pam₃CSK₄ even at a higher concentration did not overshadow that of acGM-1.8, and the two compounds might form a synergy in action. As shown in the new Figure S7c, pre-treatment of the cells with acGM-1.8 (1 µg/mL; 30 min) had no significant influence on the effect of Pam₃CSK₄, suggesting that acGM-1.8 did not antagonize with Pam₃CSK₄. In addition to these two figures, our data appearing later showed that acGM-1.8 was more related to TLR2/TLR6 than to TLR2/TRL1 (as for Pam₃CSK₄), which is in agreement with the above findings.

To answer the second question, we have corrected the x-axis in the revised manuscript. The statistics was performed at each concentration point, compared with the Pam₃CSK₄ group (circle). Two-way ANOVA with Bonferroni's multiple comparison test was performed with significance set as a **p* value < 0.05.

Q4. Speaking of the specificity of TLR agonists, the reviewer believes that cross-activation always happens, which may be a consequence of evolution. Is there any other specific TLR2 agonist either

from bacteria or fungi used for cancer research? The authors should add more discussion on this issue with new references.

A4. This is a good suggestion. We have added the relevant content in the 3rd paragraph of Discussion.

Q5. Minor issues and suggestions

a) for safety tests (fig. 6), in the groups where mice died, did they die immediately upon injection and simultaneously? When the authors show survival rate, was it calculated at the same timepoint?
b) figure legends should be improved: fig. 2 g, more explanations should be added; fig. 3 b and c, do the colored circles have the same meaning? fig. 4 4 and f, the circles/triangles should be explained. fig. 5 a has the same issue c) fig. 6 a can be drawn better. The mg/kg sign is confusing under the data column d) the carbohydrate units of glucomannan should be described.

A5. For a), for the groups receiving the highest dose of injection (100 mg/kg), we observed that the mice died within 6 h after injection, which can be considered simultaneous. When we calculated the survival rate, we did it at the same time point – 24 h after the injection of TLR agonists.

For b), we have improved the legends accordingly.

For c), we have modified the figure to avoid this confusion.

For d), the glucomannan comprises β -1,4 linked D-mannose and D-glucose monomers with a molar ratio of around 1.6:1 (mannose to glucose). We have added this information in the Materials and Methods part of the revised manuscript.

We thank Reviewer 1 for these valuable suggestions.

Reviewer #2: Expert in biomedical engineering

The manuscript reported a safe toll-like receptor agonist mimicking microbial signal to generate tumor-killing macrophages. The authors demonstrated the synthesis of acGM with different acetylation degrees and the capacity to recognise and activate macrophage TLR2 receptor with optimized acetylation degrees and structure. The key factors of material synthesis was discussed with experimental evidences and the mechanism of the tumor associated macrophages polarization was also explored. The study was comprehensive. The conclusion was supported with systematic experimental data. The technique can be very useful. I suggest that the manuscript should be published after considering the following comments.

Q1. I would suggest that the term “tumor-killing” in the title and the entire manuscript should be revised to be “tumor-suppressive”, for accuracy.

A1. We thank the reviewer for this insightful suggestion. We have changed it in the title accordingly.

Q2. Figure 2g describes the acGM dispersion change with increased pH. What will happen with decreased pH? What is the pH range for stable acGM particles? What is the stability of these acGM

particles in PBS? These will be associated with the future application of the technique, so it is good to investigate them.

A2. This is an excellent suggestion! As the reviewer suggested, we have analysed the particle size in different pH ranging from 3 to 10 (NaOH or HCl were used to adjust the pH value in PBS). We found that the particle size: 1) was stable when pH was 6-7; 2) decreased when pH exceeded 8; and 3) increased when pH fell below 5. This finding is interesting, and we speculated that the intramolecular hydrogen bond was broken when the proton concentration increased. We have added the data as a new Supplementary Figure 5b.

As the reviewer wisely pointed out, this pH-responsive feature may inspire future development of this technique into a tool for drug delivery, targeting cellular microenvironment or tissue niche where pH changes – in such cases as tumour or lysosome. We thank the reviewer for inspiring this experiment.

Q3. Beyond DLS measurement, it is good to give a size analysis as well from TEM images.

A3. We have added the data into the revised Figure 2b.

Q4. The direct binding between acGM and TLR2 is attractive. The presented co-immunoprecipitation (co-IP) data are useful, but Isothermal titration calorimetry (ITC) is mostly used to test interactions between smaller molecules including proteins. More investigation can be done to demonstrate the TLR2 binding to acGM with different modification degrees.

A4. We agree with the reviewer on the selection of the two assays. Accordingly, we have performed new experiments of co-IP using acGM with different degrees of modification (0.2, 0.6, 1.2 and 1.8). The data are shown in the revised Figure S7A. Meanwhile, as suggested, we have removed the ITC part.

Q5. In the cell biology part: did the authors see any changes of cell polarization after MDC treatment? There was probably a link between blocking endocytosis and phenotype change. How did the authors gate all CD4⁺/CD8⁺ in flow cytometry data?

A5. Firstly, we have performed a new experiment to see whether treatment of MDC will affect polarization. We pre-incubated BMDM with MDC and examined the phenotype of these cells with flow cytometry. We found no significant change and believe that MDC does not affect macrophage phenotypes (Figure-for-response-letter-only 2). Also, we have searched literature and found no report indicating such an effect of MDC, in agreement with our experiment data.

Figure-for-response-letter-only 2. FACS analysis of a typical M1 (CD11c) and M2 (CD206) marker in M0-BMDMs with or without MDC treatment.

Secondly, we have added the description of gating in the revised Supporting Information. We have added all gating illustrations in Data Source.

Q6. The authors mainly studied acGM0.2 and acGM1.8. What about acGM3.0? It looks like acGM3.0 also self-assemble to uniform nanoparticles and give good results shown in Fig. 1. Please provide an explanation.

A6. We appreciate the reviewer’s observation. We agree that acGM-3.0 also exhibits a self-assembly morphology and an effect of M1-polarization, similar to acGM-1.8 (if based on PCR data, Figure S2). We also speculated that acGM3.0 would probably have a comparable efficacy if used in the subsequent studies. Nevertheless, we selected acGM-1.8 after considering a range of factors including their nuanced effects, scientific necessity and future implications:

- 1) From the ELISA data (Figure 1 b and c) we can see that acGM-1.8 still had stronger effect than acGM-3.0 in inducing TNF- α and IL-12 (with significance), though both were good enough. Since acGM-1.8 is far better than acGM-1.2, -0.6 and -0.2 and is slightly stronger than acGM-3.0, there is no need to further increase the degree of acetylation.
- 2) After we chose acGM-1.8, we started to perform intensive in vivo experiments and mechanistic studies. For these experiments, the ethics of animal use (to use animals only necessary and, when necessary, to use minimal animal numbers) and the resource to study mechanism are important and practical concerns. Since acGM-3.0 is not superior to acGM-1.8, we believed acGM-1.8 was the most representative sample.
- 3) In future studies, we may consider further modifying acGM for different therapeutic purposes. In acGM-3.0, most if not all hydroxyl groups have been replaced by acetyl groups, making it extremely difficult for further modification. In comparison, acGM-1.8 not only has a better macrophage-regulatory activity but also offers more hydroxyl groups – thus more possibilities for further modification and functionalization.

Therefore, based on these detailed considerations, while we fully agree with the reviewer on the potential of acGM-3.0, we believe that acGM-1.8 superiorly represents the ‘active’ sample and fulfils both the scientific and practical requirements.

Q7. The authors should compare this acGM with other reported systems to strengthen the discussion. It is also good to know the authors’ views of the continual development of using acGM for anticancer therapy.

A7. We appreciate this advice. We have added relevant discussion in both the 3rd and 6th paragraph of the Discussion section in the revised manuscript. We envisage that acGM have the potential to be further developed into two types of anti-cancer therapeutic tools: 1) we will continue to evaluate the use of acGM as a direct stimulant of innate immunity for cancer immunotherapy, including optimising its formulation and tailoring its modes of administration and applying it in combination with other chemotherapeutic, immunotherapeutic or radiotherapeutic means; 2) we will also look at its potential as adjuvants, because TLR adjuvants are believed to promote cross-presentation that may maximise the power of CD8+ T cells. For these projects, we are now actively applying for research funding grants in our system and hope the data presented in this manuscript may be of help.

Q8. Some figure legends are incomplete (e.g., the safety data).

A8. We have improved the figure legends. Once again, we thank Reviewer 2 for these excellent and inspiring questions.

Reviewer #3: Expert in Cancer immunotherapy

Changing the phenotype of tumor associated macrophages towards a tumoricidal/anti-tumor immunity sustaining phenotype is an important hurdle to be taken in the immunotherapy of cancer. The work of Feng et al., therefore, is addressing an important issue. They have designed a compound which based on a number of in vitro assays and in vivo safety tests has some desirable features and maybe one of the compounds that truly helps to reach this goal. There are several issues that need to be addressed for this study to really show that their compound works as indicated by the authors.

Q1. In the abstract it is claimed that this compound – when intratumorally injected – eliminates two different tumors in vivo. This is not true, growth retardation of the tumors is shown, up to 14 days after challenge. At this point, this conclusion is not sustained by the data.

A1. We thank the reviewer for this question and would like to answer it in two aspects:

- 1) To answer both this question and Q7(3) of the reviewer (see A7 below), we have added new experiments to see what would happen if, after the 14-day observation, we stopped the treatment or continued to inject acGM. We found that from day 14 to 28, if the treatment continued, the size of S180 tumour further decreased – out of 5 samples, 2 were eliminated. Please see our completely revised Figure 4 and the relevant description.
- 2) We have changed this sentence in Abstract into ‘Intratumoral injection of acGM1.8 suppressed the growth of two tumor models in mice’ to avoid any possible overclaim.

Q2. There are two statements concerning the antitumor activity of the macrophages stimulated with this compound, however, neither was a strong antitumor effect shown in vitro (Supplemental data about 50% at best) and there is no evidence that these macrophages mediate an antitumor effect in vivo (see also below). Hence, also these statements are – at this point – not supported by the data.

A2. These are good suggestions based on which we have substantially improved the manuscript. Our points are shared below:

- 1) First, we have carefully checked throughout the manuscript and lowered the tone of our statements in several parts of the text.
- 2) We add evidence that macrophages mediate the anti-tumour effect. Thanks to the reviewer's inspirational questions Q8 and Q11, we have added new experiments such as depletion of macrophages and using nude mice (see our A8 and A11 below). The data have two implications: a) the macrophages are vital in mediating the tumour-suppressive activity because the acGM sample failed to work in the case of macrophage depletion (new Figure 5h, i and S12a); b) in addition to the crucial role of macrophages, T cells also contribute to the overall therapeutic effect; as evidenced by the data in nude mice, acGM-1.8 could also generate therapeutic effect but the effect was greatly weakened (Figure 5j, k, l and S12b). Hence, based on the data we conclude that the acGM's activity is essentially mediated by macrophages and substantially strengthened by T cells.
- 3) Our above analysis b) helps to explain why the *in vitro* data, based on the observation of the reduced viability in tumour cells stimulated by macrophage secretion, are reasonable. There is a statistically significant reduction in the viability (though, as the reviewer wisely pointed out, not drastically decreased) in the cells affected by the secretion from acGM-challenged macrophages, without the participation of T cells.

Based on these new data and analysis, we have improved both the presentation of results and statement of conclusion in the revised manuscript.

Q3. Last sentence Introduction: With anti-tumor immunity one means the adaptive immune response, the induction of anti-tumor T cells or alleviation of their effector function (restoration) but this has not been shown.

A3. First, to answer this question, along with Q8, Q10 and Q11, we have performed new experiments, such as a more comprehensive evaluation of T cell subpopulations (please see our A8 and A11 below for more details) and determination of cytokines (A10). We have now gained a better understanding of the restoration of adaptive immunity in this process to support the captioned sentence. Meanwhile, we have revised this sentence to '... TLR-2 agonist with the potential to regulate both innate and adaptive anti-cancer immune responses *in vivo*', to make the expression more accurate and lower the tone of statement.

Q4. Page 6, did the macrophages also express the well-known M2 marker CD163 next to CD06 and this they produce VEGF?

A4. The reviewer is correct! These cells, when induced into M2, did express these genes. In the revised manuscript, we have added new data showing that the BMDMs induced into an M2-phenotype expressed CD163 (FACS; revised Figure S2c) and produced VEGF (ELISA; revised Figure 1g), and acGM-1.8 could decrease the expression of these and other M2-related genes.

Q5. Figure 4: please also provide schematic drawing of the treatment procedure in this figure.

A5. We have added it as the revised Figure 4a.

Q6. Figure 4, the exact treatment is not clear, where mice injected until a certain day or until dead? And so on, a better description is required.

A6. We have added the scheme and more descriptive words. We apologise for the confusion.

Q7. Figure 4, several remarks: 1) Also provide tumor growth curves, 2) what happens when treatment stops? 3) What happens in the end, do all tumors disappear with continuous treatment? A longer observation period would be very informative with respect to the use of this compound in vivo.

A7. To answer these interesting questions, we have performed new experiments and re-organized the whole Figure 4. Our responses are below:

- 1) We have divided the tested animals into four groups, to which we: i) injected PBS for 14 days; ii) injected acGM-1.8 for 14 days; iii) injected acGM-1.8 for 14 days and stopped the treatment by injecting PBS for another 14 days; iv) injected acGM-1.8 for 14 days and continued the treatment by injecting acGM-1.8 for further 14 days. In all the groups, injection was performed every other day where applicable. As shown in our newly drawn scheme (revised Figure 4a), we inoculated S180 cells 7 days before the treatment (Day -7), waited for the tumour diameter to reach ~ 0.5 cm (Day 0) when we injected acGM or PBS. We judged a mouse 'dead' when its tumour diameter exceeded 1.5 cm and euthanised it by strictly following the animal ethics guidance.
- 2) We have provided the tumour growth curves in the revised Figure 4d for S180 and Figure 4h for b16 tumours.
- 3) We have shown that, after Day 14, if the treatment continued, the tumour size further decreased and two out of five samples were eliminated. If the treatment stopped after Day 14, the tumour size would increase. We have added the data into the revised manuscript and concluded that extended injection of acGM could effectively suppress the tumour growth.

Q8. Figure 5: The immune analyses need to be strongly improved. The reader needs to have the following information: 1) what is the increase in CD45 cells?; 2) Is there besides the increase of M1 also an increase of M2 macrophages, Ly6G+ cells and eosinophils's?; 3) do the M1 macrophages express Ly6c, hence are they inflammatory and attracted to the tumor or are they really converted TAMs?; 4) What is % of T cells within the CD45+ cell population, this needs to be established, and also heir phenotype.

A8. We are pleased to report that we have added new experiments to answer these questions and re-organized Figure 5. Our responses to each sub-question are presented below:

- 1) The proportion of CD45 cells increased (almost doubled) in acGM-treated mice (Figure 5a, PBS: 8.92%, acGM: 15.18 %).
- 2) Besides an increase in M1 macrophages, our new data revealed that there was a decrease of M2 macrophages (CD206⁺; Figure 5c. PBS: 16.9%, acGM: 12.14 %) and an increase of Ly6G⁺

cells (Figure S11b. PBS: 27.0%, acGM: 39.9%). We did not detect eosinophils (< 0.5%) in the tumour samples from both PBS and acGM treatment groups.

- 3) This is the most interesting question, and our new data provide answers in several aspects:
 - a. The IF staining for tumour slices revealed that acGM treatment markedly increased the number of M1 cells (CD11c⁺); however, most of these M1 cells were not Ly6c-positive (Figure 5m). Meanwhile, Ly6C⁺ cells were found in both acGM- and PBS-treated tumours and in a similar level (~ 15%; Figure S13a). Meanwhile, the IF staining confirmed again that there was an obvious decrease of CD206+ cells in the acGM-treated tumour (Figure 5n). This group of data suggest that inflammatory monocytes can infiltrate into the tumour in both sample and control groups, but these cells seemed not to be the target of acGM-1.8. Instead, it is more likely that acGM converted TAMs into M1 cells;
 - b. To further validate that acGM-1.8 could directly convert TAMs into M1 cells, we harvested S180 tumours from the mice, isolated macrophages, cultured them ex vivo and treated the cells with acGM-1.8. The outcomes showed that acGM-1.8 could directly increase the proportion of CD11c+ cells and decrease that of CD206 cells (Figure 5o). These data are consistent with the above findings and further suggest that acGM-1.8 modulated TAMs more than infiltrating monocytes.
- 4) Regarding the T cells. We have updated the data in the new Figure 5d-f and Figure S11a, c and d. The overall proportion of T cells in the tumor had no significant increase (CD3+, Figure S11a); however, both the number of both CD4+ (Figure 5d) and CD8+ (Figure 5e) T cells increased (both approximately 50% increase), while that of Treg population (CD4+Foxp3+, Figure 5f) decreased (more than halved). Also, the percentage of both CD4+ (13.8 to 23.0%; Figure S11c) and CD8+ T (9.7 to 17.8%; Figure S11d) cells in the blood doubled after acGM-1.8 treatment, providing further evidence of the restoration of anti-cancer adaptive immunity.

Q9. Figure 5c: quantification of the cells in several areas of the tumor and of several mice is required.

A9. We have quantified the area with corresponding fluorescent signals and divided it to area of cell nuclei (DAPI). We have added the data in Supplementary Figure 11f.

Q10. Figure 5: ELISA data, was their also production of TNF α , IL-12 and/or TGF β ? It would be nice to have this as it would connect the data with the in vitro characterization provided.

A10. We have provided the ELISA data for TNF-a, IL-12 and TGF-b, in addition to IL-10, IFN-gamma and VEGF, in the revised Figure 5g.

Q11. Last sentence page 13: this conclusion can't be made as "acGM-1.8 can efficiently generate M1-type anti-tumor macrophages and subsequently restore the T cell-mediated adaptive one against the tumor in vivo" was not shown. To establish this the authors need to: a) establish that this is the work of antitumor macrophages, by depleting macrophages or better use a mouse in which M1 macrophages can't do their job anymore (e.g. iNOS ko mice).; b) Furthermore, to show that this is not the work of T cells, they need to do depletion experiments for T cells.

A11. This is an excellent reminder! We have added new experiments in the revised manuscript to explore these two questions:

- 1) We performed *in situ* depletion of macrophages by using the liposomes of clodronate (Zeisberger et al, 2006, *British Journal of Cancer*, 95, 272). Validation of macrophage depletion is presented in Figure S12a. In the mice with macrophages depleted, the therapeutic activity of acGM-1.8 was abolished; no significant difference was observed in either the tumor size (Figure 5h) or tumor weight (Figure 5i) between the groups treated by PBS and acGM-1.8. These data suggest that macrophages play an essential role in mediating the effect of acGM.
- 2) We employed nude mice, inoculated S180 tumor, and applied acGM to see what would happen in the absence of T cells. The data showed that, in the absence of functioning T cells, acGM-1.8 could still reduce tumor size (Figure 5j) and weight (Figure 5k). However, the potency of acGM-1.8 was heavily weakened compared to that observed in the normal mice (Figure 4); the tumors kept growing, despite slower, in the treatment group.
- 3) Further analysis confirmed that, in the nude mice, acGM-1.8 could as well polarize the macrophages in the tumor from an anti-inflammatory to a pro-inflammatory phenotype (Figure 5l and Figure S12b).
- 4) These findings indicate that, for the therapeutic effect of acGM-1.8, macrophages are indispensable – they are the main cellular target and mediate the restoration of anti-tumor immune response; meanwhile, T cells also played a significant role in this action – this is agreement with the outcomes from T cell profiling and IFN- γ determination.

Q12. The discussion needs to be adapted after these essential data sets requested have been provided.

A12. We have added a new 4th and 5th paragraph of the Discussion about the newly added data in the revised manuscript.

Minor:

Q13. A careful check of the English grammar should be performed. Some words are not correct, for instance “dissembled” instead of “disassembled” and “cultural medium” instead of “culture medium” . But also some sentences are not correct. This is also observed in Legends.

A13. We have corrected these words and checked throughout the manuscript for any other typos. We apologise for the mistakes.

Q14. The figure legends of main and supplemental figures do not describe the data presented in such a way that the figures can be understood without having to look at the main text. Please provide better legends.

A14. We have improved all the figure legends in the revised manuscript. We have almost re-written all the figure legends for both main and supplementary figures to ensure that each legend is self-contained. Please see our revised version.

Q15. Page 9, line 5 instead of stating “cells” state “BMDMs” to make it more clear what you used.

A15. We have corrected this word and searched throughout the manuscript for other possible confusions.

Finally, we appreciate Reviewer 3 for all the valuable suggestions on this manuscript.

Reviewers' Comments:

Reviewer #1:

Remarks to the Author:

The authors had addressed my comments according in their revised version. The study is novel and interesting to the field of medical sciences. I feel that it will get many citations and may influence the thinking in the field.

Reviewer #2:

Remarks to the Author:

I believe that the authors have thoroughly addressed the comments with plenty of new experiments and data. The reported technology is very attractive and potentially useful. Therefore, I recommend that the manuscript should be accepted for publication in Nature Communications.

Reviewer #3:

Remarks to the Author:

I am glad to see that the authors have worked on my questions as this has certainly improved the value of their manuscript and the message their work will convey as its impact. There are a few minor things left which will ultimately complete the manuscript:

1. Figure 4a, since in group 3 the mice do not receive any injections from day 14-28, please remove the triangles below the dashed line (as they indicate injections).
2. The M&M section (supplemental) lacks a description of the low cytometry experiments, the ELISA's performed as well as the quantification of the cells stained with immunofluorescence of sections. Please add.
3. Last but not least, the authors show for two tumor models that macrophages form an important component for tumor regression. This was previously also observed in two other studies (van der Sluis Cancer Imm Res 2015; Thoreau Oncotarget 2015). It would be of interest to mention this also in the discussion of the paper.

Reviewer #3 (Remarks to the Author):

I am glad to see that the authors have worked on my questions as this has certainly improved the value of their manuscript and the message their work will convey as its impact. There are a few minor things left which will ultimately complete the manuscript:

Q1. Figure 4a, since in group 3 the mice do not receive any injections from day 14-28, please remove the triangles below the dashed line (as they indicate injections).

A1. We have corrected it accordingly. We thank the reviewer for this suggestion.

Q2. The M&M section (supplemental) lacks a description of the low cytometry experiments, the ELISA's performed as well as the quantification of the cells stained with immunofluorescence of sections. Please add.

A2. We have added the information for these experiments. We have also added a new Supplementary Figure 15 to illustrate the gating strategy for flow cytometry.

Q3. Last but not least, the authors show for two tumor models that macrophages form an important component for tumor regression. This was previously also observed in two other studies (van der Sluis Cancer Imm Res 2015; Thoreau Oncotarget 2015). It would be of interest to mention this also in the discussion of the paper.

A3. We appreciate this suggestion and have added these two interesting papers as new Ref 30 and 33 in the reference. Finally, we thank Reviewer 3 again for all the suggestions and comments.